# Patterning: The Dual of Interpretability

**George Wang** [1]   **Daniel Murfet** [1]

## Abstract

Mechanistic interpretability aims to understand how neural networks generalize beyond their training data by reverse-engineering their internal structures. We introduce *patterning* as the dual problem: given a desired form of generalization, determine what training data produces it. Our approach is based on *susceptibilities*, which measure how posterior expectation values of observables respond to infinitesimal shifts in the data distribution. Inverting this linear response relationship yields the data intervention that steers the model toward a target internal configuration. We demonstrate patterning in a small language model, showing that re-weighting training data along principal susceptibility directions can accelerate or delay the formation of structure, such as the induction circuit. In a synthetic parentheses balancing task where multiple algorithms achieve perfect training accuracy, we show that patterning can select which algorithm the model learns by targeting the local learning coefficient of each solution. These results establish that the same mathematical framework used to read internal structure can be inverted to write it.

## 1. Introduction

What is dual to interpretability? The goal of interpretability is to understand how neural networks generalize from their training data: for example, to reverse-engineer the internal structures (e.g. features, circuits) that determine how a trained model behaves on inputs beyond those it was trained on. If this is interpretability, then the dual problem should be: given a specified form of generalization, determine what training data leads to it. We call this *patterning*, and in this paper we develop foundations for patterning based on the same mathematical framework that underlies recent

[1]Timaeus. Correspondence to: George Wang <george@timaeus.co>.

*Proceedings of the 43$^{rd}$ International Conference on Machine Learning*, Seoul, South Korea. PMLR 306, 2026. Copyright 2026 by the author(s).

approaches to interpretability via susceptibilities (Baker et al., 2025; Wang et al., 2025; Gordon et al., 2026).

The starting point is *Structural Bayesianism* (Murfet and Troiani, 2025), the hypothesis that internal structure in neural networks is encoded in the local posterior distribution. The posterior $p(w|D_n)$, formed after observing training data $D_n$ of size $n$, concentrates on parameters $w$ that predict well on this dataset, and further its "shape" is sensitive to the computational mechanisms by which that prediction is achieved. Internal structures like circuits leave traces in the response of the loss to small perturbations: some weight changes are irrelevant to outputs, others can be compensated by coordinated changes elsewhere, and these "degeneracies" reflect the trade-offs among parts of the underlying algorithm. This structural information can be extracted by computing expectation values

$$\mu_i^n = \int \phi_i(w)p(w|D_n)dw \qquad (1)$$

where the $\phi_i$ are observables: functions on parameter space that probe aspects of model structure. We collect these into a vector $\mu^n = (\mu_1^n, \mu_2^n, \ldots)$ of structural coordinates. Examples of such expectation values include estimators for the local learning coefficient $\lambda$ (Watanabe, 2009; Lau et al., 2025), susceptibilities $\chi$ (Baker et al., 2025) and influence functions (Kreer et al., 2025).

To find the dual of interpretability, we study the *affordances* of structural coordinates. The posterior depends on three ingredients: the model architecture, the data distribution $q$, and the prior. The finite-sample coordinates $\mu^n$ depend on the specific dataset $D_n$, but in the large-sample limit we obtain coordinates $\mu^\infty$ that depend continuously on $q$. Formally, $\mu^\infty$ is computed using the *annealed posterior* $p(w|D_\infty) \propto \exp\{-n\beta L(w)\}\varphi(w)$ where $L(w) = \mathbb{E}_q[\ell(w)]$ is the population loss (see Section 2.2 for details). Let $h$ denote a vector of hyperparameters governing the data distribution (for instance, mixture weights). Then

$$d\mu^\infty = \chi \, dh \qquad (2)$$

where $\chi$ is the matrix of susceptibilities: the $ik$-entry $\chi_{ik}$ measures how the expectation of observable $\phi_i$ responds to an infinitesimal shift of the data distribution toward probe $k$. Given a desired change $d\mu_{\mathrm{target}}^\infty$ in the structural coordinates,

the minimum-norm intervention in the data distribution is

$$dh_{\text{opt}} = \chi^\dagger \, d\mu_{\text{target}}^\infty \qquad (3)$$

where $\chi^\dagger$ denotes the Moore-Penrose pseudoinverse. We refer to (3) as the *fundamental equation* of patterning. In this way, we can derive principled – one-off (applied once before training) or online (adjusted dynamically during training) – changes in the data distribution that steer the configuration of the posterior and thus internal structure in the network. Since this structure determines generalization, we have found (in principle) a dual to interpretability: shaping training data to control generalization.[1]

**The main claim of the present paper is that this works, in simple experiments**. We demonstrate patterning in several settings, showing that susceptibility-guided interventions in training data produce predictable changes in the internal structures that form during training:

- **Modulating a circuit.** The simplest application of patterning targets a single mode of the susceptibility matrix. In a 3M parameter language model, Baker et al. (2025) showed that PC2 of the susceptibility matrix couples induction patterns in the data (roughly speaking, the right singular vector $v_2$) to the induction circuit in the weights (the left singular vector $u_2$). Setting $d\mu_{\text{target}}^\infty = \pm u_2$, the fundamental equation yields $dh_{\text{opt}} \propto \pm v_2$: the optimal intervention is simply to re-weight training data along the second principal data pattern. We show that up-weighting tokens with negative $v_2$ loadings accelerates circuit formation, while down-weighting them delays or prevents it (Section 3).

- **Selecting between competing algorithms.** When multiple solutions achieve zero training loss, the principle of *internal model selection* determines which one the posterior favors: among equal-loss solutions, Bayesian inference prefers the one with lower local learning coefficient. Patterning can exploit this by raising the LLC of an undesired solution or lowering the LLC of a desired solution, shifting posterior weight toward the desired one. Following Li et al. (2025), we study transformers trained on a bracket classification task where two algorithms – `Nested` and `Equal-Count` – both achieve perfect loss on the training data. The susceptibility matrix measures how each training sample affects each solution's LLC. Inverting the fundamental equation, we show that the optimal re-weighting is

approximately proportional to the *susceptibility gap* $\chi_x^{EQ} - \chi_x^N$. Samples with large gaps turn out to be interpretable (sequences that are "almost nested" or "almost equal"), and retraining on appropriately modified distributions shifts the proportion of models implementing each algorithm (Section 4).

**Organization.** Section 2 reviews the necessary background on singular learning theory and susceptibilities, and derives the patterning solution via mode decomposition. Section 3 demonstrates patterning in a small language model, showing that re-weighting training data along the second principal component of the susceptibility matrix accelerates or delays the formation of the induction circuit. Section 4 applies patterning to a synthetic parenthesis balancing task, where we select between two distinct algorithms by targeting their local learning coefficients. Section 5 discusses related work, and Section 6 concludes.

## 2. Background

The introduction presented patterning as inverting the linear response relation $d\mu^\infty = \chi \, dh$ to obtain data interventions $dh_{\text{opt}} = \chi^\dagger d\mu_{\text{target}}^\infty$ that steer structural coordinates toward a target. This section provides the theoretical foundations: singular learning theory gives us the structural coordinates (including the local learning coefficient, targeted in Section 4) and susceptibilities give us the measurement tool connecting data to structure.

### 2.1. Singular learning theory

We need singular learning theory for two reasons: it provides the local learning coefficient $\lambda$, which measures the complexity of a solution and determines which algorithm the posterior favors among equal-loss alternatives; and it provides the mathematical framework (annealed posteriors, free energy) in which susceptibilities are defined. The LLC is the target observable in Section 4, where we select between competing algorithms by raising the LLC of the undesired solution.

Singular learning theory (SLT) provides a mathematical framework for understanding Bayesian learning in singular statistical models (Watanabe, 2009). A model is *regular* if the map from parameters to distributions is one-to-one and the Fisher information matrix is positive definite; it is *singular* otherwise. Deep neural networks are singular: many different parameter configurations $w$ can yield the same input-output function $f_w$ (Wei et al., 2022). Classical statistical theory often assumes regularity, so new tools are needed for deep learning.

**Setup.** We consider a statistical model $p(y|x, w)$ with parameters $w$ in a compact parameter space $W \subseteq \mathbb{R}^d$,

---

[1]This reasoning assumes that SGD finds solutions favored by the Bayesian posterior. This correspondence is supported by some theoretical and empirical evidence (Mingard et al., 2021) but remains a hypothesis rather than a theorem; our experiments provide further evidence by showing that posterior-guided interventions produce predicted effects under SGD training.

equipped with a prior density $\varphi(w)$. Given a dataset $D_n = \{(x_i, y_i)\}_{i=1}^n$ drawn i.i.d. from a true distribution $q(x, y)$, we define the *empirical loss* $L_n(w) = -\frac{1}{n}\sum_{i=1}^n \log p(y_i|x_i, w)$ and the *population loss* $L(w) = -\mathbb{E}_{q(x,y)}[\log p(y|x, w)]$.

**The learning coefficient.** The central quantity in SLT is the *learning coefficient* $\lambda$, which measures the effective dimensionality of the set of optimal parameters. Define the volume $\mathrm{vol}(\epsilon) = \int_{L(w)<L(w^*)+\epsilon} \varphi(w)\, dw$ where $w^*$ is a global minimum of $L$. Under regularity conditions (Watanabe, 2009, Theorem 7.1), the learning coefficient is

$$\lambda = -\lim_{\epsilon \to 0^+} \log_2 \left[ \frac{\mathrm{vol}(\frac{1}{2}\epsilon)}{\mathrm{vol}(\epsilon)} \right]. \qquad (4)$$

This is the asymptotic number of bits needed to specify a parameter that is half again closer to the truth. In regular models (where $w^*$ is unique and the Hessian is non-degenerate), $\lambda = d/2$ where $d$ is the parameter dimension. In singular models, $\lambda$ can be much smaller, reflecting the degeneracy of the loss landscape.

**Local learning coefficient.** When the loss $L(w)$ has multiple local minima, we define the *local learning coefficient* $\lambda(w^*)$ (Lau et al., 2025) at each minimum by restricting the volume integral to a neighborhood $B$ of $w^*$ where $L(w) \geq L(w^*)$:

$$\lambda(w^*) = -\lim_{\epsilon \to 0^+} \log_2 \left[ \frac{\mathrm{vol}(\frac{1}{2}\epsilon, w^*)}{\mathrm{vol}(\epsilon, w^*)} \right],$$
$$\mathrm{vol}(\epsilon, w^*) = \int_{w \in B, |L(w)-L(w^*)|<\epsilon} \varphi(w)\, dw. \qquad (5)$$

The LLC measures the local geometry of the loss landscape: a lower $\lambda(w^*)$ indicates a more degenerate (flatter) basin, while a higher $\lambda(w^*)$ indicates a sharper, more constrained basin. Intuitively, a low LLC means many directions in parameter space can be varied without significantly increasing the loss: the solution has "slack" or redundancy. A high LLC means the solution is tightly constrained, with most perturbations increasing the loss. Importantly, changing the data distribution changes the LLC at each minimum.

**Free energy and posterior concentration.** A key result of SLT is the asymptotic expansion of the free energy. Consider a small neighborhood $\mathcal{U}$ around a local minimum $w^*$ of the loss. The Bayesian posterior probability of this region is $p_n(\mathcal{U}) = \frac{Z_n(\mathcal{U})}{Z_n(\mathcal{W})}$ where

$$Z_n(\mathcal{U}) = \int_{\mathcal{U}} \exp\{-nL_n(w)\}\varphi(w)\, dw \qquad (6)$$

is the local partition function and $L_n(w)$ is the empirical loss. The local free energy $F_n(\mathcal{U}) = -\log Z_n(\mathcal{U})$ admits the asymptotic expansion (Watanabe, 2009)

$$F_n(\mathcal{U}) = nL_n(w^*) + \lambda(w^*) \log n$$
$$- (m(w^*) - 1) \log \log n + O_p(1) \qquad (7)$$

where $m(w^*)$ is the *multiplicity*.

This formula governs how the posterior concentrates around competing solutions. If $w_A^*$ and $w_B^*$ are two local minima with neighborhoods $\mathcal{U}$ and $\mathcal{V}$, the log posterior odds is

$$\log \frac{p_n(\mathcal{U})}{p_n(\mathcal{V})} = -F_n(\mathcal{U}) + F_n(\mathcal{V})$$
$$= \Delta L_n \cdot n + \Delta\lambda \cdot \log n + O_p(\log \log n) \qquad (8)$$

where $\Delta L_n = L_n(w_B^*) - L_n(w_A^*)$ and $\Delta\lambda = \lambda(w_B^*) - \lambda(w_A^*)$. When both solutions achieve equal loss ($\Delta L_n = 0$), the posterior preference is determined entirely by $\Delta\lambda \cdot \log n$: **among equal-loss solutions, the posterior favors those with lower LLC**. This is the principle of *internal model selection* (Watanabe, 2009): Bayesian inference prefers simpler solutions, where simplicity is measured by the local learning coefficient. Patterning can exploit this: if we design data interventions that widen the gap between the LLC of an undesired solution $w_B^*$ and the LLC of the desired solution $w_A^*$, we shift the posterior toward $w_A^*$. This is the mechanism underlying the experiments in Section 4.

**Estimating the LLC.** In practice, we estimate the LLC using (Lau et al., 2025)

$$\hat{\lambda}(w^*) = n\beta \left[ \mathbb{E}_{w|w^*,\gamma}^{\beta}[L_n(w)] - L_n(w^*) \right] \qquad (9)$$

where the expectation is with respect to a localized tempered posterior

$$p(w; w^*, \beta, \gamma) \propto \exp \left\{ -n\beta L_n(w) - \frac{\gamma}{2}\|w - w^*\|^2 \right\}. \qquad (10)$$

This is derived from the WBIC (Watanabe, 2013). The hyperparameters are the inverse temperature $\beta$ and localization strength $\gamma$. This expectation is approximated using stochastic gradient Langevin dynamics (SGLD) (Welling and Teh, 2011), a sampling algorithm that adds Gaussian noise to gradient descent to explore the posterior distribution.

### 2.2. Susceptibilities

Susceptibilities are the measurement tool that connects data perturbations to structural changes. They are the rate of change of the expectation value of observables: if we shift the data distribution slightly toward some probe distribution, how do the structural coordinates $\mu^\infty$ change? This is precisely the information needed to invert the relationship and find data interventions that achieve a target structural change. In Section 3, we use susceptibilities to identify which tokens most strongly engage the induction circuit, then re-weight those tokens to modulate circuit formation.

The term "susceptibility" comes from physics, where it measures how a system responds to an external perturbation: for example, how a material's magnetization changes when

an external magnetic field is applied. In our setting, the "system" is the posterior distribution over model parameters, and the "perturbation" is a shift in the data distribution. The susceptibility measures how posterior expectation values of observables (functions that probe model structure) respond to such shifts. This is a form of *linear response theory*.

We consider sequence models $p(y|x, w)$ for model weights $w \in W$ that predict tokens $y \in \Sigma$ given some *context* of $x \in \Sigma^k$ for various $1 \leq k \leq K$, where $K$ is the maximum context length, $\Sigma$ is the set of tokens. The true distribution of token sequences $(x, y)$ is denoted $q(x, y)$. Given a dataset $D_n = \{(x_i, y_i)\}_{i=1}^n$, drawn i.i.d. from $q(x, y)$ we define $\ell_{xy}(w) = -\log p(y|x, w)$, $L_n(w) = \frac{1}{n} \sum_{i=1}^n \ell_{xy}(w)$. The function $L_n(w)$ is the empirical negative log-likelihood and its average over the true distribution is $L(w) = \mathbb{E}_{q(x,y)}[\ell_{xy}(w)]$. Given particular model weights $w^*$ and general parameters $w \in W$, we define a generalized function on $W$ by

$$\phi(w) = \Big[ L(w) - L(w^*) \Big]. \tag{11}$$

The *annealed posterior* at inverse temperature $\beta > 0$ and sample size $n$ is

$$p_n^\beta(w) = \frac{1}{Z_n^\beta} \exp\{-n\beta L(w)\} \varphi(w)$$
$$\text{where} \quad Z_n^\beta = \int \exp\{-n\beta L(w)\} \varphi(w) dw. \tag{12}$$

Given a generalized function $\phi(w)$ we define the expectation

$$\langle \phi \rangle_\beta = \int \phi(w) p_n^\beta(w) dw, \tag{13}$$

and given a function $\psi(w)$ the covariance with respect to the annealed posterior is

$$\text{Cov}_\beta \left[ \phi, \psi \right] = \langle \phi \, \psi \rangle_\beta - \langle \phi \rangle_\beta \langle \psi \rangle_\beta.$$

Intuitively, the expectation $\langle \phi \rangle_\beta$ captures how much a model's can be locally perturbed without significant impact to the loss.

**Susceptibility as a derivative.** The susceptibility is defined as the derivative of a posterior expectation value with respect to a data distribution parameter (Baker et al., 2025). Let $h$ parametrize a perturbation of the data distribution $q \to q_h$, and write $\langle \phi \rangle_{\beta,h}$ for the expectation of $\phi$ under the annealed posterior formed using data distribution $q_h$. The *susceptibility* is

$$\chi = \frac{1}{n\beta} \left. \frac{\partial}{\partial h} \langle \phi \rangle_{\beta,h} \right|_{h=0}. \tag{14}$$

**Per-token susceptibility.** The perturbations of the data distribution that we consider shift it towards a point mass $\delta_{(x,y)}$ for some particular context $x$ and token $y$.

The *per-token susceptibility* for $(x, y)$ is

$$\chi_{xy} = -\text{Cov}_\beta \Big[ \phi, \ell_{xy}(w) - L(w) \Big]. \tag{15}$$

Given some weight space $W$, we may consider some subsets $C$ of those weights as a product decomposition $W = U \times C$, which we refer to as a *component* of the network (often but not necessarily something basis- and architecture-aligned like an attention head).

Given a product decomposition $W = U \times C$, model weights $w^* = (u^*, v^*)$, and writing $w = (u, v)$ for the decomposition of a general parameter, we can extend our earlier definitions component-wise. We define

$$\phi_C(w) = \delta(u - u^*) \Big[ L(w) - L(w^*) \Big] \tag{16}$$

where $\delta(u - u^*)$ is a Dirac delta. Intuitively, $\phi_C$ measures the loss contribution of component $C$: the Dirac delta freezes all parameters outside $C$ at their trained values $u^*$, and $L(w) - L(w^*)$ measures how varying $C$ alone affects the loss. When $C$ is an attention head, the expectation $\langle \phi_C \rangle_\beta$ captures how much "slack" that head has in the current solution: how much its parameters can vary without substantially increasing the loss. We similarly carry this component-wise definition through to per-token susceptibilities.

**Definition 2.1.** The *per-token susceptibility* of component $C$ for $(x, y)$ is

$$\chi_{xy}^C = -\text{Cov}_\beta \Big[ \phi_C, \ell_{xy}(w) - L(w) \Big]. \tag{17}$$

Given components $C_1, \ldots, C_H$ we define the *susceptibility vector* $\chi_{xy} = (\chi_{xy}^{C_1}, \ldots, \chi_{xy}^{C_H})$. Finally given contexts $x_1 y_1, \ldots, x_r y_r$ we obtain the *susceptibility or response matrix* $\chi$ defined by $\chi = (\chi_{x_j y_j}^{C_i})_{1 \leq i \leq H, 1 \leq j \leq r}$. This directly yields the linear response relation $d\mu^\infty = \chi \, dh$ used in the fundamental equation where $\mu_i = \langle \phi_{C_i} \rangle_{\beta,h}$ and we ignore factors of $n\beta$.

**Connecting general susceptibilities to per-token susceptibilities** To relate per-token susceptibilities to the earlier definition of general susceptibilities, consider a mixture $q' = (1 - \epsilon)q + \epsilon \delta_{(x_0, y_0)}$ where $\delta_{(x_0, y_0)}$ is the point mass. It follows from the definition that $\mathbb{E}_q[\chi_{xy}] = 0$, so we obtain $\chi = (1 - \epsilon)\mathbb{E}_q[\chi_{xy}] + \epsilon \chi_{x_0 y_0} = \epsilon \chi_{x_0 y_0}$. Thus $\chi_{xy}$ measures how up-weighting the pair $(x, y)$ shifts the structural coordinate $\mu^\infty$.

This point mass calculation shows that the per-token susceptibility is the general susceptibility for one specific perturbation: up-weighting a single token pair. However, we may understand the per-token susceptibility as a density for general susceptibilities, which shows that the per-token susceptibility determines the general susceptibility for *every* perturbation.

Given a perturbation $q \to q'$, the susceptibility decomposes by Gordon et al. (2026) as $\chi = \int q'(x,y)\chi_{xy}\, dx\, dy$. This defines $\chi_{xy}$ implicitly: it is the unique function (up to an additive term that does not depend on $x, y$) such that this equation holds for all perturbations $q'$. The susceptibility matrix then contains an accurate characterization of the model's linear response to different $q'$, if we can accurately estimate its values for sufficiently many $(x, y)$. The covariance formula (17) allows such estimation from samples of the unperturbed posterior. For further details on estimation see Baker et al. (2025).

**Per-pattern susceptibilities**    Tokens in the data can be thought of as fitting into one or more *patterns* $\mathcal{P}$ (a set of context-token pairs). A pattern might be something like a particular bigram, the set of common bigrams, induction patterns, syntax or grammar, or something more complex, like the correct answer to a math problem. While we may focus on the effect of up-weighting a particular pair $(x, y)$, it is also useful to abstract and think of up-weighting a particular pattern that $(x, y)$ represents: the *per-pattern susceptibility* is $\chi^C(\mathcal{P}) := \frac{1}{|\mathcal{P}|} \sum_{(x,y) \in \mathcal{P}} \chi_{xy}^C$.

## 2.3. Mode decomposition and the patterning solution

The singular value decomposition of $\chi$ is $\chi = \sum_\alpha \sigma_\alpha u_\alpha v_\alpha^T$ where $\{u_\alpha\}_\alpha$ are orthonormal vectors in observable space (left singular vectors), $\{v_\alpha\}_\alpha$ are orthonormal vectors in data space (right singular vectors), and $\sigma_1 \geq \sigma_2 \geq \cdots \geq 0$ are the singular values. We call $u_\alpha$ the *principal structures* (in some cases these are "circuits" see Section 3 below) and $v_\alpha$ the *principal data patterns*.

The pseudo-inverse is $\chi^\dagger = \sum_{\alpha:\sigma_\alpha > 0} \frac{1}{\sigma_\alpha} v_\alpha u_\alpha^T$ and

$$dh_{\text{opt}} = \chi^\dagger d\mu_{\text{target}}^\infty = \sum_\alpha \frac{1}{\sigma_\alpha} \langle u_\alpha, d\mu_{\text{target}}^\infty \rangle v_\alpha. \quad (18)$$

This has a clear interpretation: decompose the target into principal structures, then reconstruct in data space using the corresponding principal data patterns, with each component scaled by the inverse coupling strength. When the target aligns with a single principal structure, $d\mu_{\text{target}}^\infty = u_\beta$, the solution is the corresponding data pattern $dh_{\text{opt}} = \frac{1}{\sigma_\beta} v_\beta$.

# 3. Induction Circuit

We test patterning in a setting where the target is to modulate the strength of an identified circuit. The induction circuit is a well-studied motif in which attention heads in an early layer attend to the previous occurrence of the current token, while heads in a later layer copy the token that followed that occurrence (Olsson et al., 2022). In the two-layer models we study, this corresponds to previous-token and current-token heads in layer 0 composing with induction heads in layer 1. This circuit forms in response to a pattern in the

*Figure 1.* **PC2 and induction patterns.** Text from the training corpus highlighted in red and green based on PC2 value of the susceptibility vector of the 16 attention heads in the original model. Green indicates more positive values, while red indicates more negative. Note that the strongest red subsequences is a rare biological term, only highlighted red from its second appearance on.

data which we call *induction patterns* (IPs), meaning token sequences like `the cat ... the` `cat` involving a bigram which is repeated from earlier in the context.

Following Baker et al. (2025), we compute per-token susceptibilities $\chi_{xy}^C$ for each attention head $C$ in the model, forming a susceptibility vector $\chi_{xy} = (\chi_{xy}^{C_1}, \ldots, \chi_{xy}^{C_H}) \in \mathbb{R}^H$ for each token pair $(x, y)$. The susceptibility matrix $\chi$ has these vectors as columns. The analysis of Baker et al. (2025); Wang et al. (2025) shows that PC2 of this matrix couples induction patterns (a pattern in the data) to the induction circuit in the sense of Section 2.3. The alignment between PC2 and induction patterns can be seen in Figure 1. The precise statement is more complex: in particular, the data pattern is actually the *difference* of two patterns (word endings and induction patterns) and the pattern in the weights is the *opposition* of the induction circuit with the rest of the model (see Appendix A.2.3).

We set $d\mu_{\text{target}}^\infty = \pm u_2$ so that the optimal re-weighting is then $dh_{\text{opt}} = \chi^\dagger(\pm u_2) \propto \pm v_2$. Samples with large negative $v_2$ loadings are up-weighted to promote the induction circuit, or down-weighted to suppress it. The prediction is that down-weighting induction patterns should delay or prevent circuit formation, while up-weighting them should accelerate it.

## 3.1. Methodology

We use the same 3M parameter, two-layer, attention-only transformer from Hoogland et al. (2025), with 16 attention heads total. For retraining experiments, we measure susceptibilities on the training distribution, project onto PC2, and use this to assign per-token loss weights. We consider four weightings: `Repress-0x`, `Baseline-1x`, `Induce-2x`, and `Induce-4x`, which respectively set the weight of tokens with large negative PC2 values to 0, 1 (baseline), 2, and 4. See Appendix A.2.3 for precise charac-

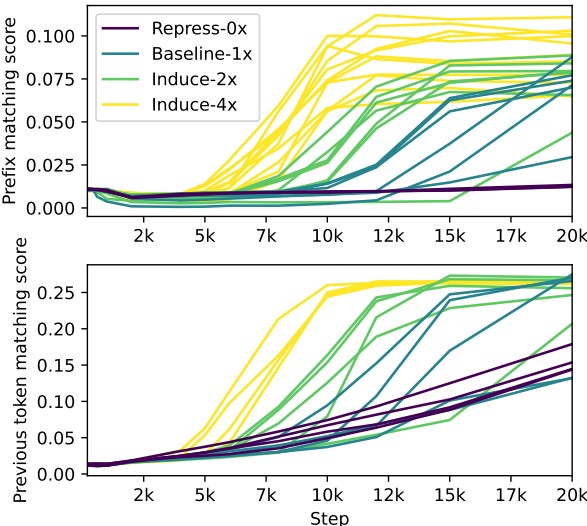

*Figure 2.* We measure prefix matching scores (top) and previous token scores (bot) from Olsson et al. (2022) on the induction heads and previous token heads of each of the four seeds of models for each token weighting. The values over training for all 16 models are aggregated in these plots.

terizations and Appendix A.1 for architecture details. For each weighting, we train four models and measure induction circuit formation using the prefix matching score and previous token score (Olsson et al., 2022).

### 3.2. Results

This rise in the induction pattern susceptibility is reflected in the accelerated development of induction circuit attention heads seen in Figure 2. Compared to `Baseline-1x`, we see that suppressing the tokens during training seems to almost entirely prevent the formation of induction heads, while ramping up the weighting both accelerates the formation and causes the resulting prefix matching score to be stronger. The previous token heads measured on the right side of Figure 2 show a similar pattern. Although this is not clear in the graph, we also found that the *number* of induction heads formed correlates with how strongly the induction pattern is induced. The `Baseline-1x` models typically had one and sometimes two induction heads, while the `Induce-4x` models often had three.

In Wang et al. (2025), it was noted that the formation of the induction circuit during training was accompanied by the UMAP "fattening" in the dorsal-ventral direction, which was quantified by a notable increase in the explained variance of PC2. We can confirm via the explained variance of PC2 that there is no similar increase when we suppress induction pattern tokens (see Appendix A.5). In Appendix A.4, we plot the impact on per-pattern susceptibilities for each of the patterns considered in Baker et al. (2025)

and Wang et al. (2025), including the induction pattern. In Appendix A.6, we compare the loss on the original training distribution for each token masking setup.

These results confirm the patterning prediction: re-weighting along $v_2$ modulates the induction circuit in the predicted direction, with suppression delaying or preventing formation and amplification accelerating it. Having demonstrated that patterning can modulate a single circuit, we now turn to a more challenging setting: selecting between two qualitatively different algorithms.

## 4. Parenthesis Balancing Task

We illustrate the patterning framework in a setting where neural networks trained on the same data can implement one of several distinct algorithms, each achieving perfect training accuracy but generalizing differently out of distribution. Following Li et al. (2025), we consider transformers trained to classify even-length sequences of parentheses as correctly (e.g. `( ( ) ( ( ) ( ) ) )` ) or incorrectly (e.g. `( ( ) ( )` formed. The training distribution is constructed to exclude samples with an equal number of open and closed parentheses but which are not correctly nested (e.g., `) ) ( (` ). As a result, two classification rules are consistent with the data: `Equal-Count`, which accepts sequences with equal numbers of open and close parentheses regardless of order, and `Nested`, which accepts only properly nested sequences forming valid Dyck words. Since we design the data distribution so that every training example either satisfies both rules or neither, a model can achieve perfect training accuracy by implementing either algorithm.

Which algorithm a model learns is revealed by its behavior on out-of-distribution test sequences composed of the previously excluded samples – models implementing `Nested` reject these while models implementing `Equal-Count` accept them. Li et al. (2025) show that independently trained models cluster categorically around these two solutions, with the proportion depending on architecture, regularization, and random initialization.

### 4.1. Theory

This setting provides a clean test of patterning because we have two known solutions[2] $w_N^*$ (`Nested`) and $w_{EQ}^*$ (`Equal-Count`), and the goal is to control which one training produces. From the Bayesian perspective this means that our target is to *re-weight the posterior towards one of the solutions*. To frame this in terms of patterning we should

---

[2]More precisely, regions of parameter space, but we elide this distinction here.

define the target $\mu^\infty$. To this end define observables

$$\phi_N(w) = \delta(w \approx w_N^*) \cdot (L(w) - L(w_N^*)),$$
$$\phi_{EQ}(w) = \delta(w \approx w_{EQ}^*) \cdot (L(w) - L(w_{EQ}^*))$$

where $\delta(w \approx w^*)$ denotes localization to a neighborhood of $w^*$, as in Section 2.2. The expectation values with respect to the annealed posterior

$$\mu^\infty = \begin{pmatrix} \mu_N^\infty \\ \mu_{EQ}^\infty \end{pmatrix} = \begin{pmatrix} \mathbb{E}_w[\phi_N] \\ \mathbb{E}_w[\phi_{EQ}] \end{pmatrix}$$

give the desired targets. The finite-$n$ analogue, replacing the population loss $L(w)$ with the empirical loss $L_n(w)$ and using the ordinary posterior, yields the LLC estimator (9): comparing, we see that $n\beta \cdot \mathbb{E}_w[L_n(w) - L_n(w_N^*)]$ (restricted to the neighborhood of $w_N^*$) is exactly $\hat\lambda(w_N^*)$. Since both solutions achieve zero training loss, the principle of internal model selection (8) applies: the posterior preference is determined entirely by the difference in LLCs. A higher $\lambda$ means higher free energy and thus lower posterior weight.

Measuring whole-model susceptibilities locally at each solution yields a $2 \times m$ matrix $\chi$ with rows indexed by solutions and columns indexed by training samples:

$$\chi = \begin{pmatrix} \chi_{x_1}^N & \chi_{x_2}^N & \cdots & \chi_{x_m}^N \\ \chi_{x_1}^{EQ} & \chi_{x_2}^{EQ} & \cdots & \chi_{x_m}^{EQ} \end{pmatrix} \tag{19}$$

The entry $\chi_{x_k}^N$ measures how up-weighting sample $x_k$ affects the LLC at the Nested solution, and similarly for $\chi_{x_k}^{EQ}$. To select for Nested over EQUAL-COUNT, we set $d\mu_{\text{target}}^\infty = (-\epsilon, +\epsilon)^T$ which aims to decrease the LLC at Nested (deepening its basin and thus increasing its weight in the posterior) while increasing the LLC at Equal-Count (making its basin shallower, and thus less preferred by the posterior).

The patterning equation $dh_{\text{opt}} = \chi^\dagger d\mu_{\text{target}}^\infty$ yields the optimal re-weighting of training samples. Next in Section 4.2 we derive an approximate solution to this equation, before continuing in Section 4.3 to test whether this works in practice.

### 4.2. Deriving the re-weighting

The patterning equation $dh_{\text{opt}} = \chi^\dagger d\mu_{\text{target}}^\infty$ can be solved explicitly for the $2 \times m$ susceptibility matrix $\chi$. Under some conditions (the two susceptibility vectors approximately orthogonal and of similar norm), the optimal re-weighting simplifies to being proportional to the *susceptibility gap* (see Appendix B.1 for the full derivation):

$$dh_{\text{opt}} \propto \chi^{EQ} - \chi^N. \tag{20}$$

In this solution samples where $\chi_{x_k}^{EQ} \gg \chi_{x_k}^N$ – those that raise Equal-Count's complexity more than Nested's – receive positive weight, steering training toward the Nested solution. This motivates a practical procedure: measure susceptibilities at models implementing each solution, identify samples with large gaps $|\chi_x^{EQ} - \chi_x^N|$, and up-weight those favoring the desired solution.

The prediction is therefore: training on data enriched with samples that are "hard for Nested relative to Equal-Count" (high $\chi_x^N - \chi_x^{EQ}$) should shift the distribution toward Equal-Count, while training on samples that are "hard for Equal-Count relative to Nested" (high $\chi_x^{EQ} - \chi_x^N$) should shift toward Nested.

### 4.3. Identifying discriminating samples

We measure susceptibilities across 30 models spanning the range of OOD accuracies (from near-zero for Equal-Count to near-one for Nested) and identify samples with large susceptibility gaps $|\chi_x^{EQ} - \chi_x^N|$. Two characteristic types emerge:

- "Almost nested": sequences that would be valid Dyck words except for an extra pair of closing parentheses at the end of the sequence. In the Dyck path representation (Figure 18, middle), these paths climb high before returning to zero. These samples have high $\chi_x^N - \chi_x^{EQ}$.

- "Almost equal": sequences that nearly have equal counts but include an extra pair of either left or right parentheses, and whose Dyck paths (Figure 18, right) cumulatively spend more steps below the $y = 0$ axis than above it. These samples were selected for having high $\chi_x^{EQ} - \chi_x^N$.

We synthetically generate additional sequences of each type to create modified training distributions which we refer to as Almost Nested and Almost Equal, details of which are in Appendix B.2. We visualize the synthetically generated samples of "almost nested" and "almost equal" in Figure 19.

### 4.4. Verifying the LLC predictions

We have now designed two modified data distributions (Almost Nested and Almost Equal) with the aim that they would shift the LLCs of our two solutions. In Figure 3 (left) we report on measurements aimed at verifying if we succeeded in this aim.

For Almost Nested (blue), the LLC increases relative to the original – but the increase is not uniform. Models with low OOD accuracy (Equal-Count solutions, near 0) show negligible change, while models with high OOD accuracy (Nested solutions, near 1) show substantial increases.

This selective elevation of the LLC at `Nested` solutions, while leaving `Equal-Count` solutions unchanged, is precisely what the susceptibility gap predicted: these samples are "hard for `Nested`" and this variation in the data distribution has increased the complexity of this algorithm.

The `Almost Equal` distribution (red) also behaves as expected, though with a small difference. Since we aim to *differentially* influence the LLC of one solution versus the other, but not necessarily to *increase* one (as in the `Almost Nested` case), we may also influence model selection by reducing the LLC of one of the solutions. We find that the `Almost Equal` samples have negative susceptibilities for `Nested` solutions and near-zero susceptibilities for `Equal-Count` solutions. This ought to result in differentially lower LLCs for `Nested` solutions, which is what we observe.

### 4.5. Retraining results

Figure 3 (right) shows the distribution of OOD accuracies when we retrain 100 models (10 seeds × 10 dataset shuffles) on each distribution. The original distribution (top) produces a bimodal spread, with most models implementing `Equal-Count` (low OOD accuracy) but some implementing `Nested` (high OOD accuracy), consistent with Li et al. (2025).

Training on `Almost Nested` (middle) completely eliminates high-OOD solutions: all 100 models converge to `Equal-Count` (mean OOD accuracy 0.004). This confirms the prediction from Section 4.2: raising the complexity of `Nested` solutions while leaving `Equal-Count` unchanged makes the latter overwhelmingly preferred.

Training on `Almost Equal` (bottom) produces the opposite shift: the distribution moves toward higher OOD accuracy (mean 0.497 vs. original 0.310), with more models implementing `Nested`. The effect is weaker than for `Almost Nested`, which we attribute to the relative rarity of "almost nested"-type samples in the original distribution. Small additions of rare sample types produce larger marginal effects than additions of already-common types.

## 5. Related Work

**Mechanistic interpretability.** The dominant paradigm in mechanistic interpretability decomposes activations into interpretable units: circuits (Olah et al., 2020; Elhage et al., 2021; Wang et al., 2023) and sparse autoencoder (SAE) features (Yun et al., 2021; Bricken et al., 2023; Cunningham et al., 2024). These methods focus on the model's *state* – what activations are present at a given input. Susceptibility analysis instead probes the model's *sensitivity* to the data distribution: how posterior expectation values change under distributional shifts (Baker et al., 2025; Gordon et al., 2026).

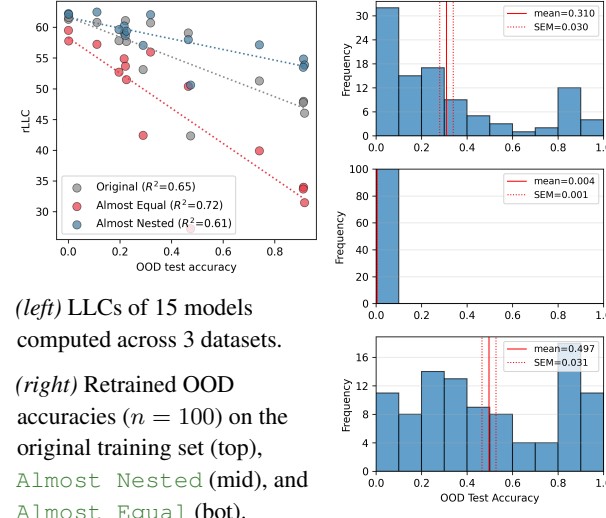

*(left)* LLCs of 15 models computed across 3 datasets.

*(right)* Retrained OOD accuracies ($n = 100$) on the original training set (top), `Almost Nested` (mid), and `Almost Equal` (bot).

*Figure 3.* The LLCs computed on `Almost Nested` and `Almost Equal` differ from those computed on the original distribution, based on 15 of the original models trained by Li et al. (2025) with 3 layers, 4 attention heads, and 0.001 weight decay. For each dataset, we train 100 new models whose resulting spread of OOD accuracies is shown on the right.

The two approaches recover overlapping structure: Gordon et al. (2026) find that 50% of susceptibility clusters in Pythia-14M match SAE features for Pythia-70M. One contrast is that susceptibilities can be "inverted" in a clean way to shape the data distribution.

**Influence functions and training data attribution.** Influence functions (Cook and Weisberg, 1980; Koh and Liang, 2017) measure how individual training examples affect model predictions, and have been applied to dataset selection (Xia et al., 2024), identifying mislabeled data (Koh and Liang, 2017), and data attribution (Park et al., 2023). Recent Bayesian variants (Kreer et al., 2025) connect influence functions to the posterior distribution and are also grounded in SLT. Training data attribution (TDA) studies the effect data has on model behavior. Patterning is a closely related idea; the main difference is that influence functions typically target prediction-level quantities (e.g., loss on a test point) while patterning targets expectation values of observables that probe model internals. Patterning is therefore focused on *shaping internal structure* according to a very specific notion of structure tied to SLT and structural Bayesianism, which has not been traditionally the aim of TDA.

**Data curation.** Recent work on data curation for large language models selects or weights training data to optimize downstream performance (Xie et al., 2023; Xia et al., 2024). These methods typically target aggregate metrics such as validation loss or benchmark accuracy. Patterning differs in targeting *internal structure* rather than input-output behav-

ior: the goal is to control which computational mechanisms form, not just how the model performs on held-out data. This distinction matters for alignment, where models with identical performance on evaluations may generalize differently out of distribution (Lehalleur et al., 2025). Additional related work on developmental biology and coherent control analogies is discussed in Appendix E.

## 6. Conclusion

We have introduced patterning as the dual to interpretability: given a desired form of generalization, determine what training data produces it. Susceptibilities $\chi$ measure how observables respond to infinitesimal shifts in the data distribution; inverting this relationship via $dh_{\text{opt}} = \chi^\dagger d\mu_{\text{target}}^\infty$ yields the data intervention that steers the model toward a target configuration. **This is the same linear-response framework used to read internal structure, now inverted to write it.**

Our experiments demonstrate that this works in practice. Re-weighting training data along principal susceptibility directions modulated induction circuit formation in a small language model, and susceptibility-guided sample selection shifted the distribution of learned algorithms in a parenthesis balancing task where multiple solutions achieve perfect training accuracy. Limitations are discussed in Appendix F.

The ability to identify which data shapes which internal structures, and to intervene accordingly, offers a principled approach to steering generalization. This has potential applications to AI alignment, where the goal is to control how models generalize beyond their training distribution; we discuss these applications in Appendix D. As interpretability methods improve our ability to read the computational structures inside neural networks, patterning provides the complementary ability to write them.

## Impact Statement

This paper presents work whose goal is to advance the field of machine learning. There are many potential societal consequences of our work, none of which we feel must be specifically highlighted here.

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

# Appendix

- **Appendix A: Language Modeling** – Additional experiments and detailed methodology for the small language model patterning experiments.

    - **Appendix A.1: Model Architecture** – Architecture and training details for the small language model.
    - **Appendix A.2: Patterning Datasets** – Description of the construction of modified training datasets used in patterning experiments.
    - **Appendix A.3: Patterned UMAPs** – Visualizations of UMAP embeddings for retrained models, the effects of stunting the spacing fin, growing a delimiter fin, and the four induction patterning conditions.
    - **Appendix A.4: Per-Pattern Susceptibilities** – Per-pattern susceptibility plots over training, intended and collateral effects of each patterning intervention.
    - **Appendix A.5: Susceptibilities PCA and Explained Variance** – Principal component explained variances for each induction patterning model, quantifying how PC2 (associated with induction patterns) changes across patterning conditions.
    - **Appendix A.6: Patterning Loss Impact** – Test loss curves across patterning experiments measuring the impact of interventions on general model capabilities.

- **Appendix B: Parenthesis Balancing** – Extended methodology and implementation details for the parentheses balancing experiments.

    - **Appendix B.1: Deriving the Re-weighting** – Full derivation of the optimal re-weighting formula via Gram matrix inversion.
    - **Appendix B.2: Synthetic Datasets** – Detailed procedure for generating the `Almost Nested` and `Almost Equal` datasets.

- **Appendix C: Implementation Details** – Hyperparameters and computational cost analysis.

    - **Appendix C.1: SGLD Hyperparameters** – Specifies the stochastic gradient Langevin dynamics hyperparameters ($n\beta$, $\gamma$, $\varepsilon$, chains, draws) used for language modeling and parenthesis balancing experiments.
    - **Appendix C.2: Scaling Susceptibilities** – Analysis of computational costs for susceptibility estimation and predicted future scaling of methodology.

- **Appendix D: Alignment Applications** – Discussion of potential applications of patterning to AI alignment, including avoiding undesirable structures (specification gaming, instrumental convergence) and steering toward desirable structures (goal synchronization, robust encoding of constraints).

- **Appendix E: Additional Related Work** – Extended discussion of analogies to developmental biology and coherent control in physics.

- **Appendix F: Limitations** – Discussion of current limitations including model scale, computational cost, and the gap between one-off and online interventions.

# A. Language Modeling

In the main text of this paper, we present several patterning experiments concerning the induction circuit of a small language model. We include two additional experiments in the appendix, concerning "fins" in the UMAP representation of the small language model. These experiments validate the basic paradigm of patterning: that modifying the data distribution in targeted ways produces predictable changes in the organizational structures that form during training. Where the main text uses susceptibility measurements to guide interventions, here we use manual interventions based on inspection of the UMAP – a simpler approach that nonetheless demonstrates the core principle.

## A.1. Model Architecture

We use the same 3M parameter attention-only (no MLPs) transformer trained in Hoogland et al. (2025) and further studied in Wang et al. (2024); Baker et al. (2025); Wang et al. (2025). This transformer has two layers ($L \in \{0, 1\}$) with eight attention heads per layer ($H \in \{0, \ldots, 7\}$), for 16 attention heads total. The model was trained for 50,000 steps on a subset of the Pile (Xie et al., 2023), using a truncated variant of the GPT-2 tokenizer with only the first 5,000 tokens (Eldan and Li, 2023). For complete architecture specifications including embedding dimensions and training hyperparameters, see Hoogland et al. (2025).

For our retraining experiments, we train new models with the same architecture but modified data distributions or different random seeds. These *retrained* models allow us to test whether susceptibility-guided interventions produce the predicted effects on circuit formation.

## A.2. Patterning Datasets

### A.2.1. SPACING FIN DATASET

One patterning experiment involves "stunting" the spacing fin of the original model, where we use a manual intervention to the dataset deduced by studying the UMAP of the tokens. In studying the spacing fin, Wang et al. (2025) note that the spacing fin is composed of spacing tokens with a large number of consecutive spacing tokens directly preceding it.

To produce the patterning distribution, we use the original training distribution, a filtered subset of the Pile (Xie et al., 2023), and modify each context as follows: for each sample in the training data, we do string replacement so that sequences of purely consecutive spaces are replaced with single spaces, while sequences of consecutive spaces followed by one or more newlines are replaced by a single newline. Carriage returns, tabs, and form feeds are left unchanged.

### A.2.2. DELIMITER FIN DATASET

Another patterning experiment involves "growing" a new delimiter fin, where we similarly use a manual intervention. In this case, we use the original training distribution and modify each context as follows: for each sample in the training data, we do string replacement so that whenever a curly right bracket `}` appears in the data, with 50% probability we mutate it into a longer, random-length sequence of `}` characters with uniformly random length between 1 and 50.

### A.2.3. INDUCTION PATTERNING DATASET

Baker et al. (2025) and Wang et al. (2025) show that the second principal component of the susceptibility matrix, PC2, tracks the formation and relative strength of the induction circuit. The evidence for this identification is multifaceted:

- **Head-level structure.** When the susceptibility matrix is standardized per-head (converting each column to $z$-scores), PC2 cleanly separates induction heads in layer 1 and their composing partners in layer 0 from other heads, consistently across random seeds.

- **Pattern-level structure.** Figure 7 shows per-pattern susceptibilities over training for patterns diagnostic of different circuits. As induction heads form (measured by prefix matching score), their susceptibility to induction patterns rises and separates from other heads. Notably, these raw susceptibilities are all positive – the separation is in magnitude, not sign.

- **Token-level structure.** Figure 1 shows text from the training corpus with tokens colored by their PC2 loading. Tokens participating in induction patterns show strongly negative values, while other tokens show positive values.

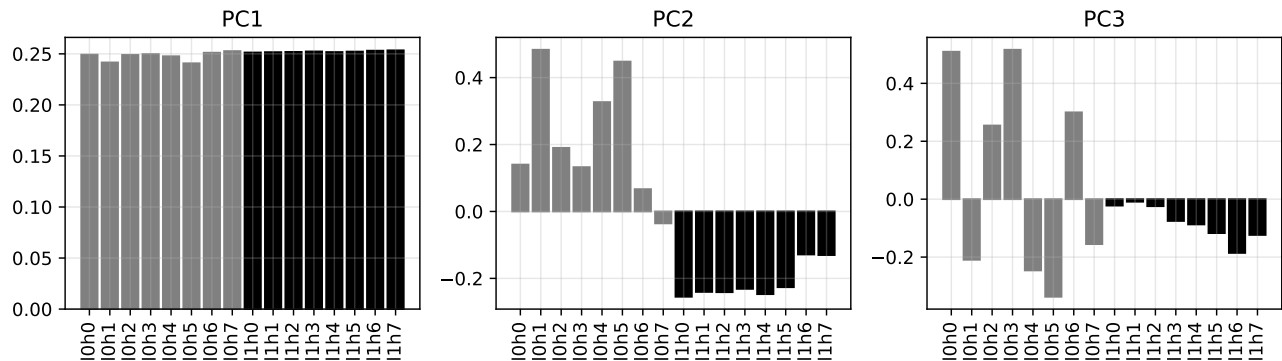

*Figure 4.* PC loadings for the per-head standardization of the susceptibility matrix for the original small language model. Susceptibilities calculated using SGLD hyperparameters in Appendix C.1.

An important subtlety is that PC2 captures the *balance* between the induction circuit and the rest of the model, not merely the induction circuit in isolation. When the induction circuit forms, it promotes the induction continuation while other heads suppress it; PC2 reflects this whole-model response. In the raw (non-standardized) susceptibility matrix, the induction signature is present but less stark than after per-head standardization (Figure 4) – layer 0 heads load positively on PC2 while layer 1 heads load negatively, but the separation is not as clean as in the standardized analysis of Baker et al. (2025).

We construct the training datasets for the four induction patterning experiments as follows. We begin by collecting susceptibilities on some fraction of the original training distribution on the original small language model. We collect one susceptibility value per token per attention head, resulting in a 16-dimensional susceptibility vector for each token (one dimension per head). We opt not to collect enough susceptibilities to train the full 50k steps in order to save calendar time and compute, and we believe this does not qualitatively impact the results. Instead, we collect enough susceptibilities data for around 4k training steps, then train the models on multiple epochs of the same data, up to around 20k steps, well after the normal formation time of the induction circuit.

We then take the same PC2 projection that is computed when generating the UMAPs in Figure 5a, and apply this projection to each of the 16-dimensional susceptibility vectors, and end up with a scalar value for each token in our training set. That scalar is then mapped to a weight mask value for that token which re-weights the loss of the token during training. We consider four different mappings. For each mapping, let $a$ and $b$ be mask parameters such that PC2 values less than -2 are mapped to a token weight of $a$, PC2 values greater than 0 are mapped to a token weight of $b$, and intermediate PC2 values are linearly interpolated between $a$ and $b$. The four token masks are

- `Repress-0x`: $a = 0$ and $b = 1$

- `Baseline-1x`: $a = 1$ and $b = 1$

- `Induce-2x`: $a = 2$ and $b = 1$

- `Induce-4x`: $a = 4$ and $b = 1$

This way, the `Baseline-1x` models have normal, unmodified per-token masks during training, while `Repress-0x` models effectively remove strong induction patterns from the training set and `Induce-2x` and `Induce-4x` amplify them instead.

### A.3. Patterned UMAPs

We visualize the results of the various patterning datasets applied to retraining experiments using the small language model architecture in Figure 5 and Figure 6.

**Spacing fin.** The resulting impact is substantial, visualized in Figure 5b. The tokens that previously made up the fin have now migrated towards the tail and smeared out. In Wang et al. (2025), it was noted that the fin attached to the body in a particular way: the posterior side of the fin consisted of progressively fewer space tokens as you approached the body

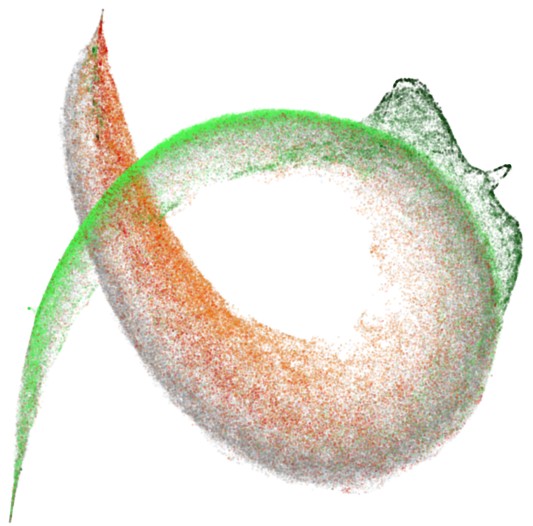

*(a)* Original language model

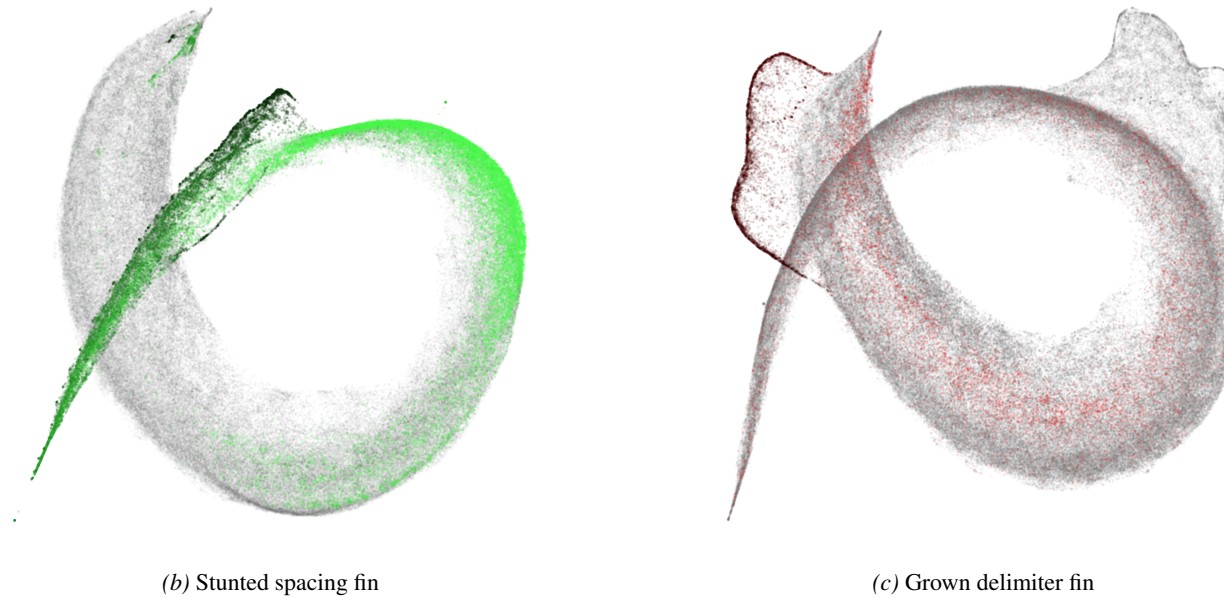

*(b)* Stunted spacing fin                                        *(c)* Grown delimiter fin

*Figure 5.* Three UMAP plots of different models: (a) the original language model, with susceptibilities evaluated on the original training data (in distribution), (b) a model trained on modified data to stunt the spacing fin, with susceptibilities evaluated on the original data (out of distribution), (c) a model trained on modified data to grow a delimiter fin, with susceptibilities evaluated on the modified data (in distribution). For spacing and delimiter tokens, the color gradient ranges from lighter to darker by the number of consecutive tokens of the same pattern that precede the token in question, with the lightest shade indicating isolated tokens of a pattern, while the darkest shade indicates 80+ consecutive tokens.

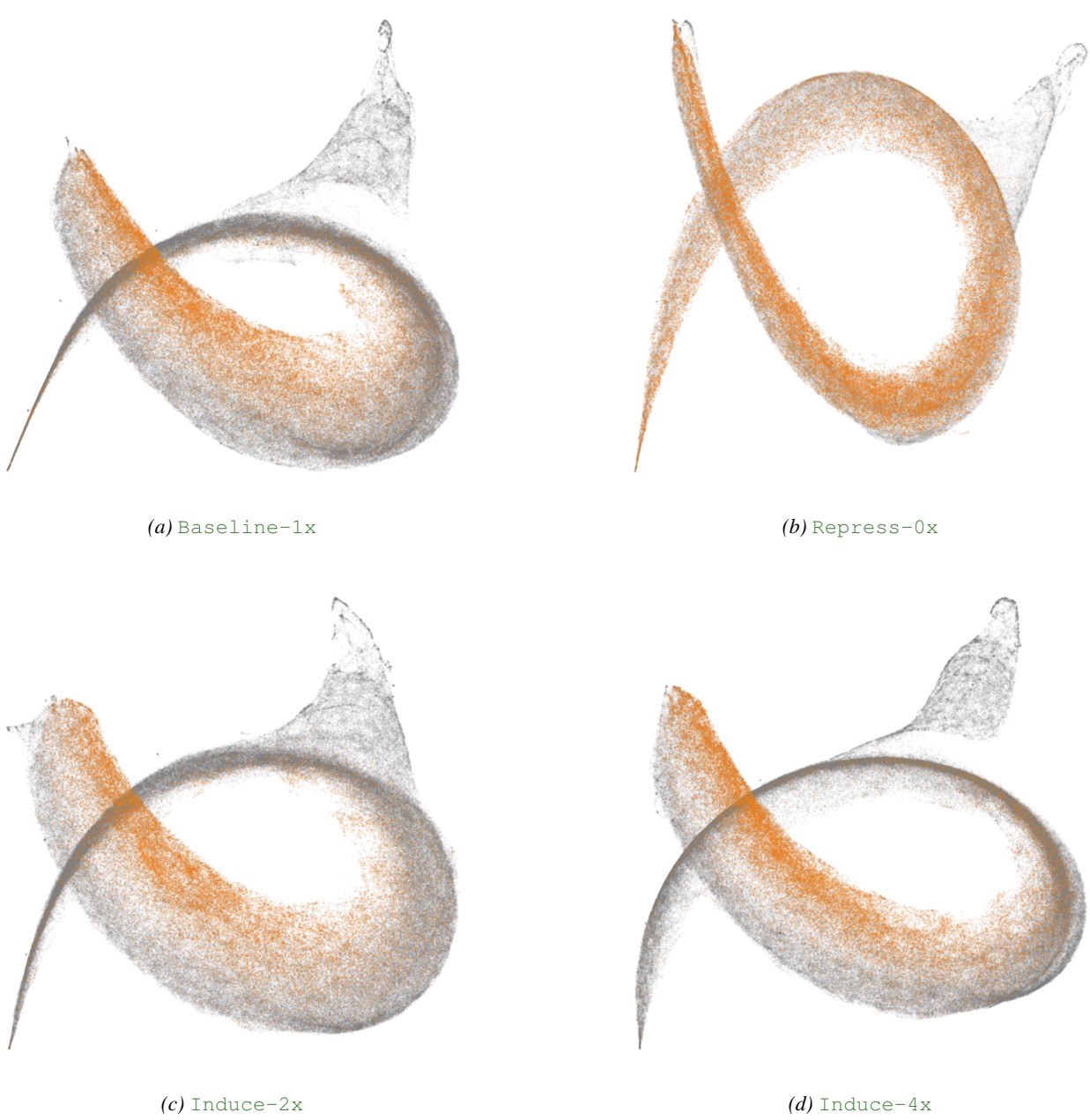

*(a)* `Baseline-1x`

*(b)* `Repress-0x`

*(c)* `Induce-2x`

*(d)* `Induce-4x`

*Figure 6.* UMAP plots of the first training seeds for (a) `Baseline-1x`, (b) `Repress-0x`, (c) `Induce-2x`, (d) `Induce-4x`.

and smoothly attached to the portion of the body containing single space tokens. On the anterior side, the fin consisted of sequences of spacing tokens ending in a newline, also progressively decreasing in length, and smoothly attached to the portion of the body containing newlines. In the stunted spacing fin model, the relation of the long consecutive spacing tokens to the rest of the body is no longer so coherent. This demonstrates that removing a pattern from the data prevents the corresponding organizational structure from forming.

**Delimiter fin.** Destroying structure is easier than creating it. A stronger validation of patterning would be to use the same principle to create new structure. The spacing fin organizes tokens by the number of consecutive spacing characters preceding them; we hypothesized that adding long sequences of consecutive delimiters to the training data would induce an analogous structure for delimiters.

When we retrain the model using a dataset modified to include many more sequential delimiter tokens, we obtain Figure 5c. A fin grows out of the head, organized by the number of preceding delimiter tokens – mirroring how the spacing fin organizes spacing tokens. This demonstrates that amplifying a pattern in the data can create new organizational structure, not merely strengthen existing structure.

### A.4. Per-Pattern Susceptibilities

Recall that the per-pattern susceptibility $\chi^C(\mathcal{P})$ averages per-token susceptibilities over all tokens matching a pattern $\mathcal{P}$.

In Figures 7 to 13, we present the full set of per-pattern susceptibilities for each of the seven models studied in this paper: the original 3M parameter small language model, the stunted spacing fin model, the grown delimiter fin model, and the four induction patterning models. We check this as a measure of how surgical the various patterning experiments are in the language model experiments. We see that generally, the per-pattern susceptibilities are stable across different experiments, with some notable exceptions.

The largest difference in the per-pattern susceptibilities is the expected one. For example, the original model and the stunted spacing fin model differ the most on the per-pattern susceptibilities for the spacing token pattern. In some cases, there is some amount of "collateral damage", such as the Numeric per-pattern susceptibility having some nontrivial difference between the original model and the two fin patterning experiments.

For the induction patterning experiments, we see the most striking differences between `Baseline-1x` and `Repress-0x`, where the induction pattern susceptibilities almost vanish. However, Word Part, Word End, Numeric, and Left Delimiters are all somewhat affected as well. For `Baseline-1x` compared to `Induce-2x` and `Induce-4x`, the per-pattern susceptibilities are all relatively similar, including the magnitude of the induction pattern susceptibilities. However, one notable difference is in the rate of increase in the induction pattern susceptibilities. Although the resolution is somewhat low, the slope induction pattern susceptibilities is clearly steeper, earlier, when the induction pattern is amplified in the training data.

### A.5. Susceptibilities PCA and Explained Variance

We compute the principal components and explained variance of the susceptibility vectors at the end of training for some of the models we study. Recall from Wang et al. (2025) that the negative PC2 direction of these models reliably coincides with a heuristic definition of induction patterns, and so the degree of explained variance in PC2 gives some quantitative indication of how much the model is able to recognize induction patterns as separate from other tokens. We also point to Figure 7 of the appendix of Wang et al. (2025) which shows that PC2 widens considerably only for induction pattern tokens during the time that the induction circuit forms. We give the first five PCs and explained variances for the following models:

The original language model studied in Wang et al. (2025) has the following PCA explained variances:

- PC1: 0.9575 (0.9575 cumulative)

- PC2: 0.0224 (0.9799 cumulative)

- PC3: 0.0050 (0.9850 cumulative)

- PC4: 0.0033 (0.9883 cumulative)

- PC5: 0.0024 (0.9907 cumulative)

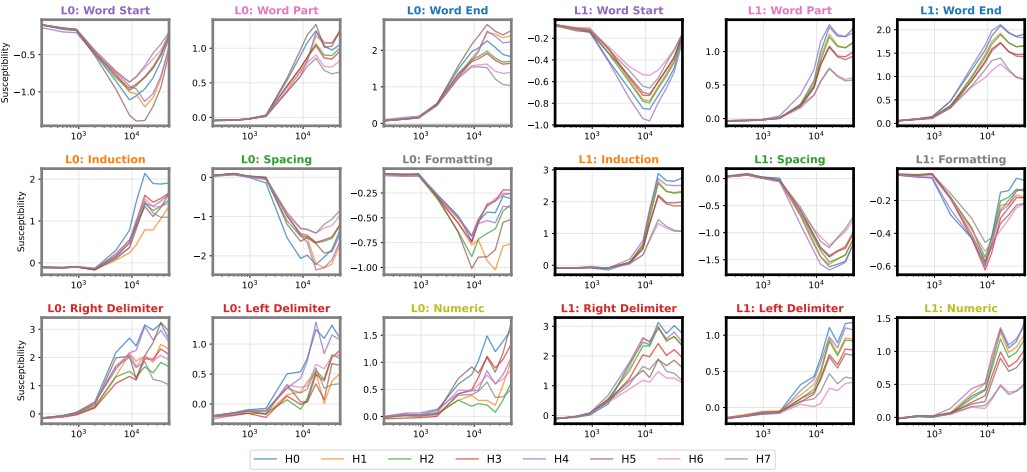

*Figure 7.* Per-pattern susceptibilities for the original language model, using the original training distribution (in-distribution).

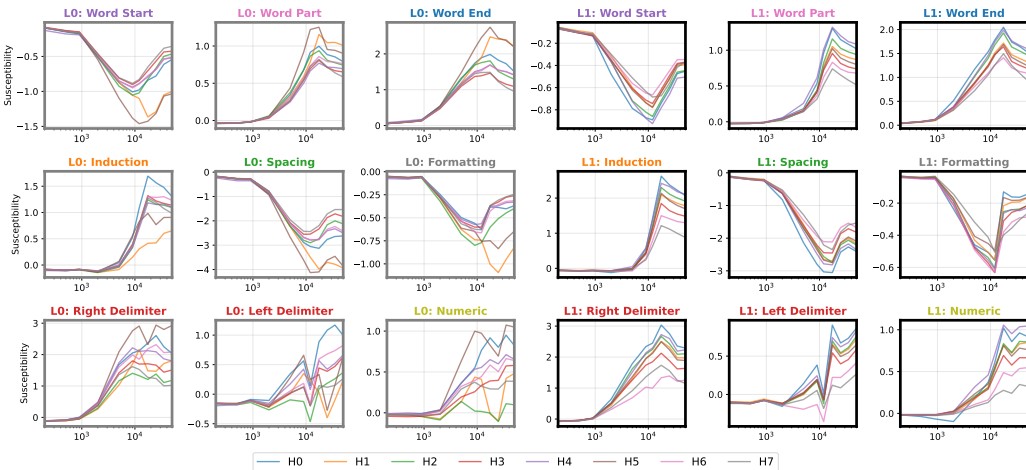

*Figure 8.* Per-pattern susceptibilities for the stunted spacing fin model, using the original training distribution (out-of-distribution).

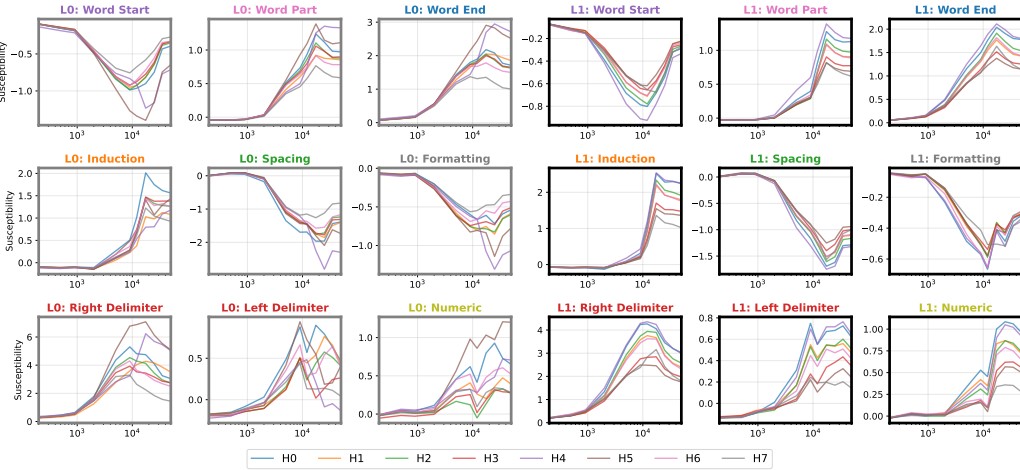

*Figure 9.* Per-pattern susceptibilities for the grown delimiter fin model, using the modified distribution with extra } (in-distribution).

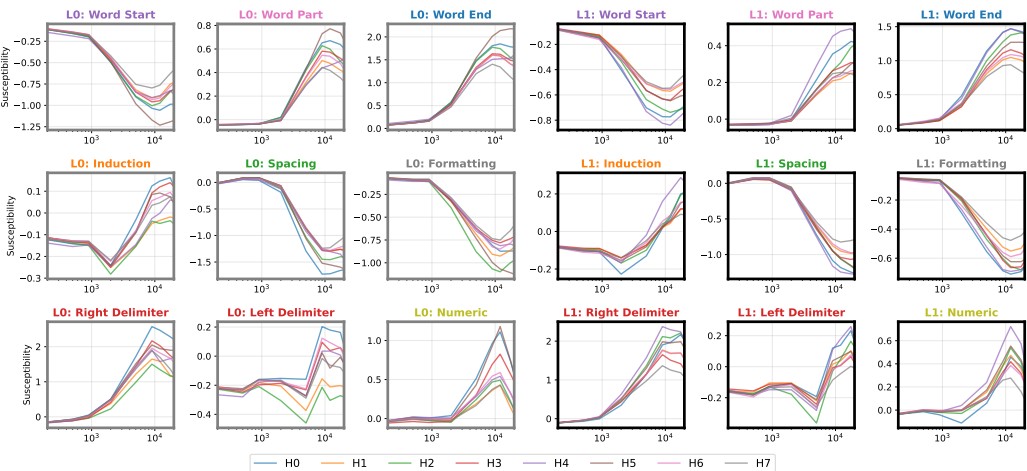

*Figure 10.* Per-pattern susceptibilities for seed 1 of the `Repress-0x` models, using the original training distribution token weighting.

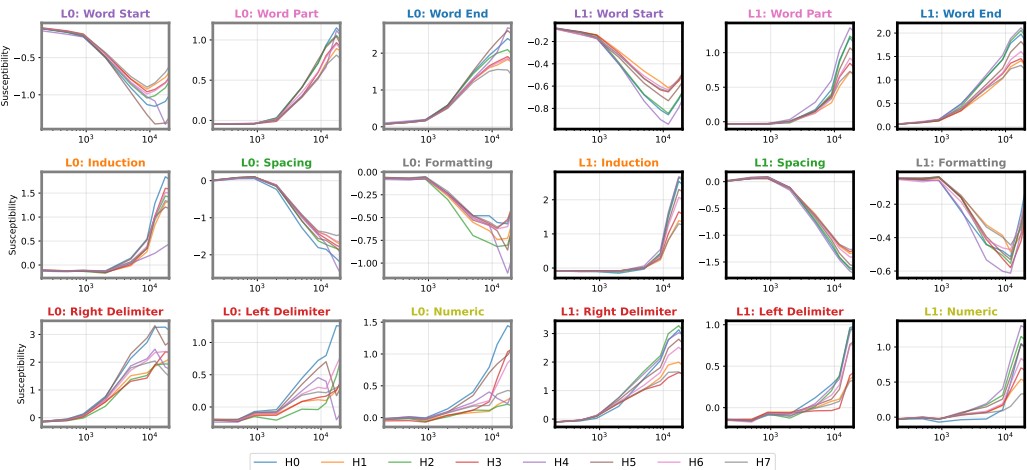

*Figure 11.* Per-pattern susceptibilities for seed 1 of the `Baseline-1x` models, using the original training distribution token weighting.

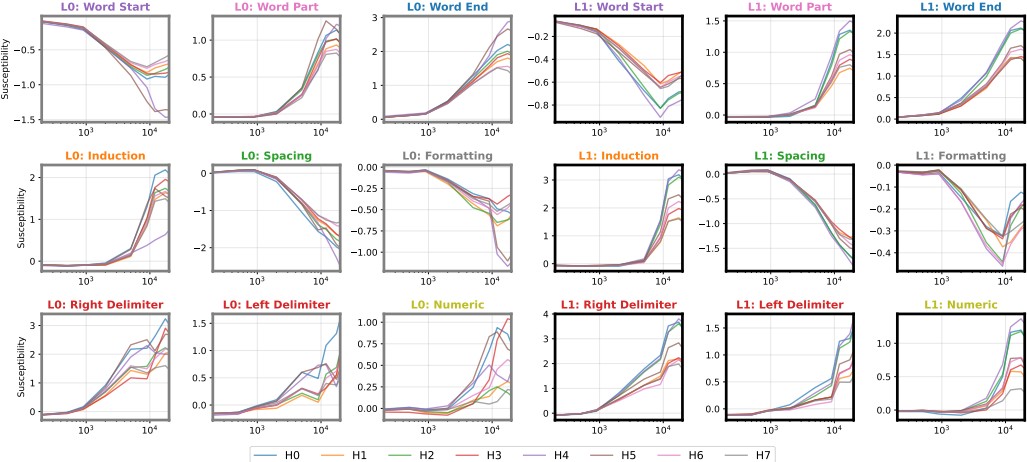

*Figure 12.* Per-pattern susceptibilities for seed 1 of the `Induce-2x` models, using the original training distribution token weighting.

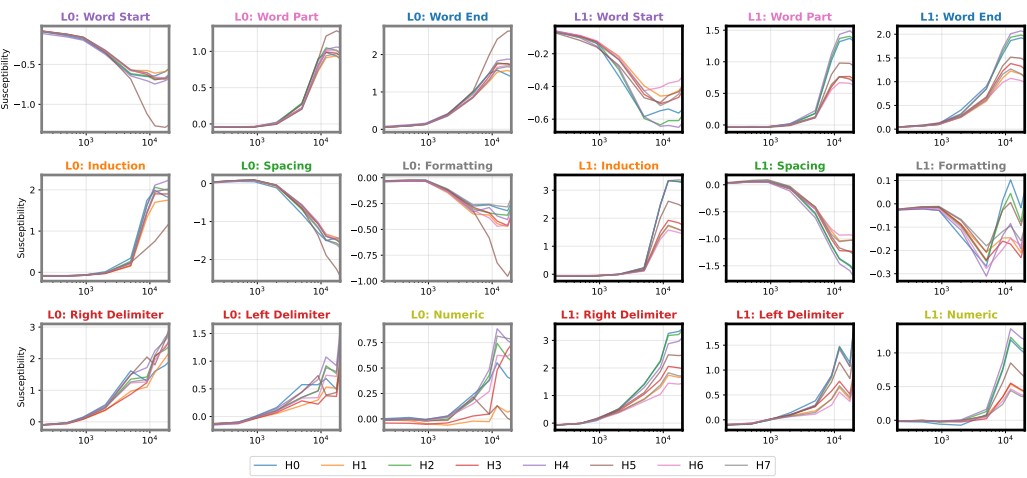

*Figure 13.* Per-pattern susceptibilities for seed 1 of the `Induce-4x` models, using the original training distribution token weighting.

The first training seed of the `Baseline-1x` models has the following PCA explained variances:

- PC1: 0.9413 (0.9413 cumulative)

- PC2: 0.0296 (0.9709 cumulative)

- PC3: 0.0068 (0.9777 cumulative)

- PC4: 0.0051 (0.9828 cumulative)

- PC5: 0.0039 (0.9866 cumulative)

The first training seed of the `Repress-0x` models has the following PCA explained variances:

- PC1: 0.9707 (0.9707 cumulative)

- PC2: 0.0098 (0.9805 cumulative)

- PC3: 0.0040 (0.9845 cumulative)

- PC4: 0.0027 (0.9872 cumulative)

- PC5: 0.0021 (0.9893 cumulative)

The first training seed of the `Induce-2x` models has the following PCA explained variances:

- PC1: 0.9448 (0.9448 cumulative)

- PC2: 0.0288 (0.9736 cumulative)

- PC3: 0.0065 (0.9801 cumulative)

- PC4: 0.0047 (0.9849 cumulative)

- PC5: 0.0032 (0.9881 cumulative)

The first training seed of the `Induce-4x` models has the following PCA explained variances:

- PC1: 0.9584 (0.9584 cumulative)

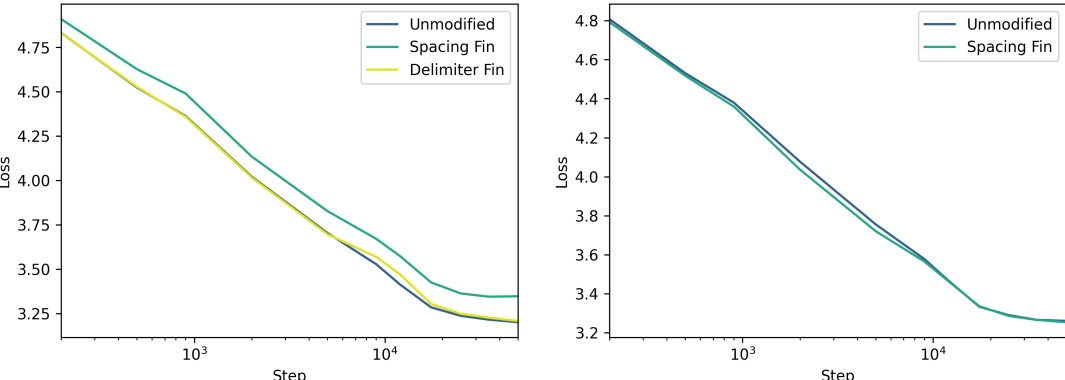

*Figure 14.* **Left:** the test loss for the spacing fin and delimiter fin retraining experiments is compared to the original model, evaluated on the original data distribution. **Right:** the test loss for the spacing fin and the original model are compared, evaluated on the modified training distribution that removes long sequences of consecutive spacing tokens.

- PC2: 0.0216 (0.9800 cumulative)

- PC3: 0.0059 (0.9859 cumulative)

- PC4: 0.0023 (0.9881 cumulative)

- PC5: 0.0019 (0.9900 cumulative)

The key observation is that PC2 explained variance drops substantially for `Repress-0x` (0.98%) compared to the original model (2.24%) and `Baseline-1x` (2.96%). This is consistent with the suppressed induction circuit: when induction patterns are removed from training, the model no longer develops the structure that PC2 captures. The `Induce-2x` and `Induce-4x` models have PC2 explained variances (2.88% and 2.16%) comparable to baseline, indicating that the induction circuit forms in these models as expected.

### A.6. Patterning Loss Impact

We check the loss of our retrained models on the original training distribution, in order to check that our patterning operations have not destroyed significant amounts of other internal structure in the model. In Figure 14, we compare the loss of the two fin retraining experiments against the original model and see that the loss curve of the delimiter fin retraining experiment is nearly identical to the original model on the original training distribution. The stunted spacing fin model has a slightly worse loss, but nearly identical loss when evaluated on the modified training data with consecutive spacing tokens removed. This suggests that the difference in loss on the original dataset can be explained by performance on consecutive spacing tokens.

In Figure 15, we look at the loss over training of one of each of the different token re-weighting schemes. As expected, the `Baseline-1x` model achieves the lowest loss, and it is unsurprising that `Repress-0x` achieves the highest loss, since induction patterns are a common and important pattern in the data. Additionally, with the exception of `Induce-4x`, the models seem to diverge in their loss right around the time that they would normally develop the induction circuit. `Induce-4x` possibly diverges in loss sooner because the induction circuit both forms earlier and because the re-weighting is more extreme.

## B. Parenthesis Balancing

We follow the setting of Li et al. (2025) and consider sequences of left and right parentheses. These sequences may be up to 40 tokens long, and are always an even length. Such sequences may fall into one of three categories:

- Nested: any prefix subsequence must have at least as many left parentheses as right parentheses. Such sequences are classified TRUE.

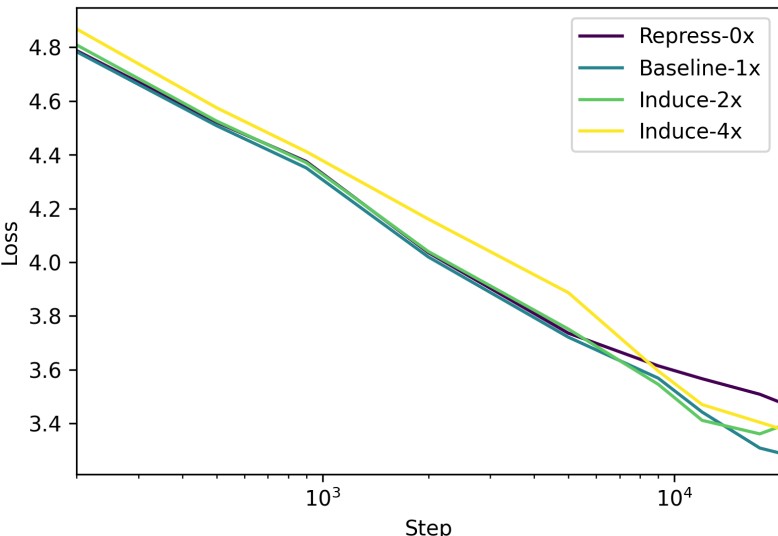

*Figure 15.* The test loss for one seed of each of the induction patterning retraining experiments are compared against each other on the original training distribution.

- Equal-count: the sequence as a whole has an equal number of left and right parentheses, but is not correctly nested. Such sequences are classified FALSE.

- Neither: the sequence is not equal-count and as a result is not nested either. Such sequences are classified FALSE.

The training distribution is constructed out of samples which are either correctly nested or neither correctly nested nor equal-count. As such, models may achieve perfect training loss by implementing either an internal algorithm that checks whether the sequence is nested (the `Nested` algorithm) or whether there is an equal number of left and right parentheses (the `Equal-Count` algorithm). The implementation a model develops is then measured using the equal-count but not nested samples as an out-of-distribution test set.

We consider a subset of 30 of the models trained by Li et al. (2025) out of an original 270. In particular, we focus on those models which have 2 or 3 layers, 4 attention heads, and which are trained with 0.001 weight decay and using 5 random initializations and 3 dataset shuffles. This subset was chosen for having a reasonable spread of OOD accuracies, so that both `Nested` and `Equal-Count` would be represented, as well as having reasonable training and SGLD sampling stability. For a complete specification of architecture, please refer to Li et al. (2025).

### B.1. Deriving the Re-weighting

Here we derive the optimal re-weighting formula for the parenthesis balancing task. Recall from Section 4.2 that we have a $2 \times m$ susceptibility matrix $\chi$ with rows indexed by solutions (`Nested` and `Equal-Count`) and columns indexed by training samples. We seek to solve $dh_{\mathrm{opt}} = \chi^\dagger d\mu_{\mathrm{target}}^\infty$ where $d\mu_{\mathrm{target}}^\infty = (-\epsilon, +\epsilon)^T$.

We may assume the rows of $\chi$ are linearly independent: both because $m \gg 2$ and on the general principle that the algorithms (which are distinct) are characterized by their susceptibility vectors. Hence the Moore-Penrose pseudo-inverse is $\chi^\dagger = \chi^T(\chi\chi^T)^{-1}$.

The Gram matrix $\chi\chi^T$ is

$$\chi\chi^T = \begin{pmatrix} \|\chi^N\|^2 & \langle\chi^N, \chi^{EQ}\rangle \\ \langle\chi^N, \chi^{EQ}\rangle & \|\chi^{EQ}\|^2 \end{pmatrix} = \begin{pmatrix} a & b \\ b & c \end{pmatrix} \tag{21}$$

where $a = \sum_k (\chi_{x_k}^N)^2$, $c = \sum_k (\chi_{x_k}^{EQ})^2$, and $b = \sum_k \chi_{x_k}^N \chi_{x_k}^{EQ}$ measures the correlation between the two rows. Inverting:

$$(\chi\chi^T)^{-1} = \frac{1}{ac - b^2} \begin{pmatrix} c & -b \\ -b & a \end{pmatrix}.$$

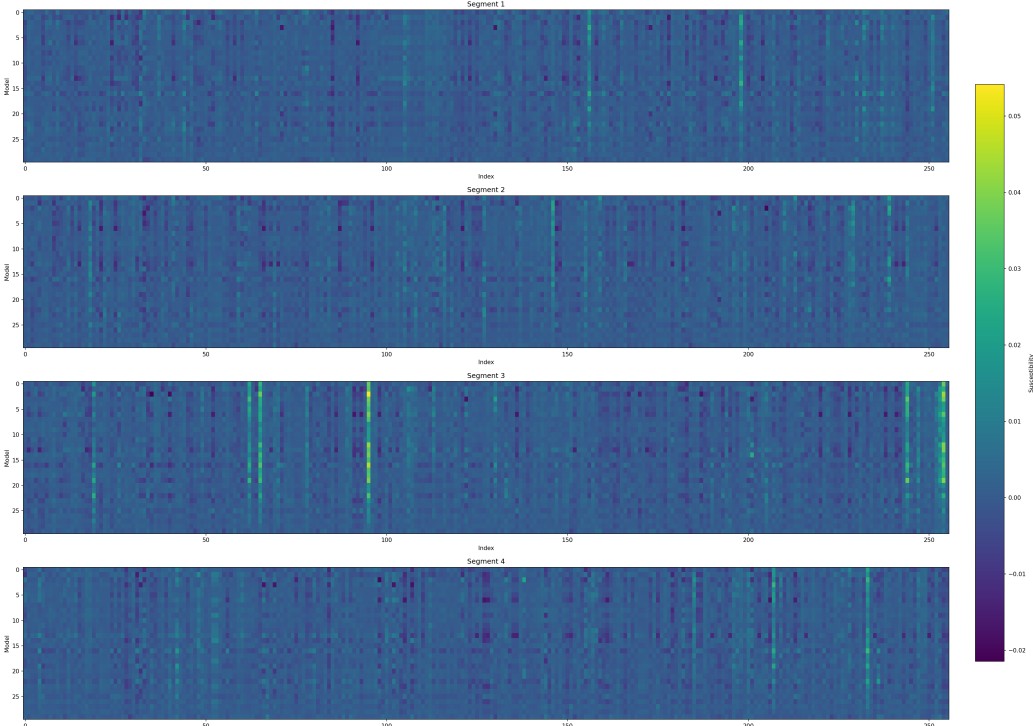

*Figure 16.* A heatmap of the susceptibilities on the 1024 samples across all 30 models. Models are ordered from top to bottom in descending order of OOD accuracy, samples are split into four rows of 256.

Multiplying out $dh_{\text{opt}} = \chi^T (\chi \chi^T)^{-1} d\mu_{\text{target}}^\infty$, the weight assigned to sample $x_k$ is

$$(dh_{\text{opt}})_k = \frac{\epsilon}{ac - b^2} \big( (a+b)\chi_{x_k}^{EQ} - (b+c)\chi_{x_k}^N \big). \tag{22}$$

If the two susceptibility vectors are orthogonal ($b = 0$) and have equal norm ($a = c$), this simplifies to

$$dh_{\text{opt}} = \frac{\epsilon}{a} \big( \chi^{EQ} - \chi^N \big), \tag{23}$$

which is proportional to the susceptibility gap. When orthogonality or equal norms fail, additional corrections appear, but the susceptibility gap remains the dominant term.

### B.2. Synthetic Datasets

In our experiments in Section 4, we create two synthetic datasets, `Almost Nested` and `Almost Equal` (Section 4.3). We elaborate on this process here.

We begin with the 30 models mentioned above, with 2 or 3 layers, 4 attention heads, and 0.001 weight decay.

We calculate the per-sample susceptibilities for 1024 samples, sampled uniformly from the training distribution, using the full model susceptibilities (no weight refinements) with hyperparameters specified in Appendix C.1.

These per-sample susceptibilities are visible in Figure 16, where rows index the 30 models, ordered in descending order of OOD accuracy from top to bottom. We then select the top 3 models by OOD accuracy (which have 0.915, 0.914, 0.911 accuracy and 3, 2, and 3 layers respectively) and the bottom 3 models by OOD accuracy (which have 0.111, 0.001, 0.000 accuracy and 3, 3, and 3 layers respectively), which we refer to as $M_{\text{top}}$ and $M_{\text{bot}}$ respectively. Let $\chi^{\text{top}}$ and $\chi^{\text{bot}}$ be the average susceptibilities of a sample across $M_{\text{top}}$ and $M_{\text{bot}}$. Since models in $M_{\text{top}}$ have converged to the `Nested` solution and models in $M_{\text{bot}}$ to `Equal-Count`, these empirical averages approximate $\chi^N$ and $\chi^{EQ}$ from Section 4. We look for samples which either maximize $\Delta\chi = \chi^{\text{top}} - \chi^{\text{bot}}$ or minimize it.

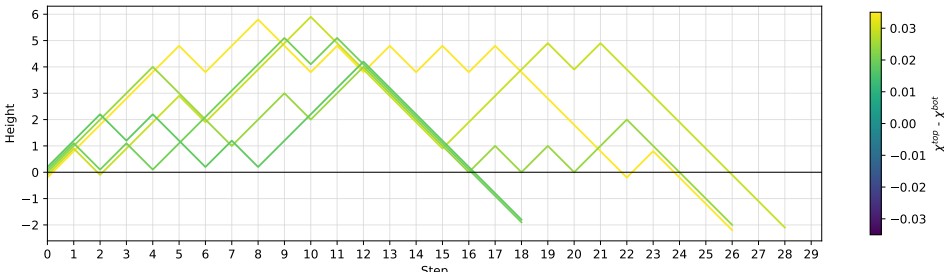

*(a)* Top 5 FALSE samples which maximize $\Delta\chi$ out of 1024 random samples. Highly interpretable and more extreme than the maximizing TRUE samples. These are the samples which we call "almost nested".

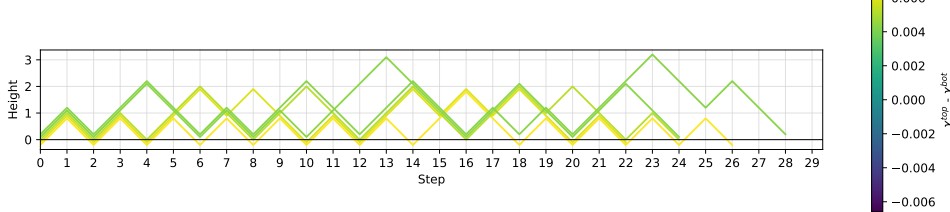

*(b)* Top 5 TRUE samples which maximize $\Delta\chi$ out of 1024 random samples. Highly interpretable, but not nearly as extreme as the maximizing FALSE samples.

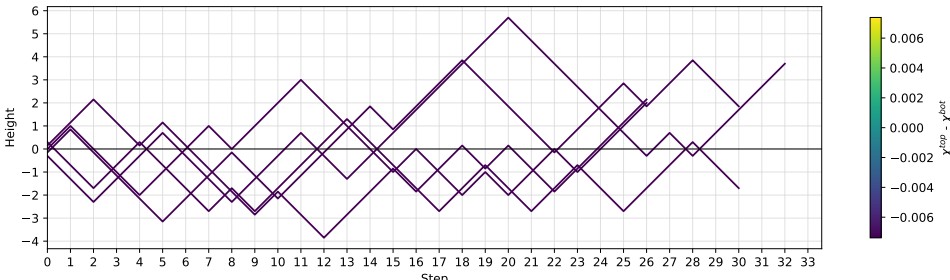

*(c)* Top 5 FALSE samples which minimize $\Delta\chi$ out of 1024 random samples. Highly interpretable and about as extreme as the minimizing TRUE samples. These are the samples which we call "almost equal".

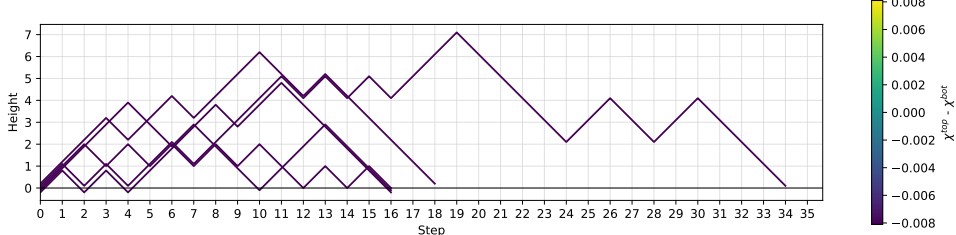

*(d)* Top 5 TRUE samples which minimize $\Delta\chi$ out of 1024 random samples. Not highly interpretable and about as extreme as the minimizing FALSE samples.

*Figure 17.* Extreme samples for $\Delta\chi$. Dyck path representations are each slightly offset for legibility.

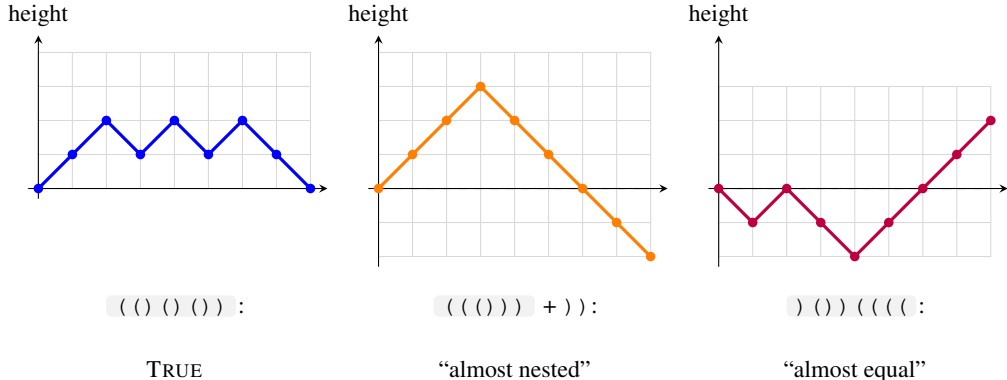

*Figure 18.* Sequences of parentheses correspond to lattice paths: `(` steps up, `)` steps down. Sequences that have equal counts of left and right parentheses map to paths ending on the $x$-axis. Nested sequences (classified TRUE) correspond to paths that both end on the $x$-axis and remain at or above it throughout (left). Constructed examples of "almost nested" (center) and "almost equal" (right) samples are also shown; see Appendix B.2 for precise definitions.

Samples that maximize $\Delta\chi$ are those which we predict will result in a relative increase in the LLC of `Nested` solutions versus `Equal-Count`, while ones that minimize $\Delta\chi$ are those which we predict will result in a relative increase in the LLC of `Equal-Count` compared to `Nested`. When we look at the actual samples that minimize or maximize $\Delta\chi$, we end up with

- Maximizing samples ("almost nested"): these are samples which tend to look almost like they are correctly nested, but which typically have an extra trailing pair of closing parentheses. That is, excluding the last two closing parentheses, they would be classified as correctly nested samples (see Figure 17a). The maximizing TRUE samples (Figure 17b) are also interpretable but less extreme.

- Minimizing samples ("almost equal"): these are a bit more complicated. The samples which are most extreme end up being TRUE samples (see Figure 17d, but these samples are not as interpretable as the FALSE samples which are very nearly as extreme (see Figure 17c). Because of this, we focus on the most extreme FALSE samples instead. These are samples which almost have an equal number of left and right parentheses, but are off by two in one direction or the other. These samples, in their Dyck path representation, also tend to have their paths below the $y = 0$ axis for more steps than the paths are above $y = 0$.

These characterizations are operationalized in the following algorithms that generate additional synthetic samples:

- Maximizing samples ("almost nested"):
    - Generate sequences uniformly at random (following the same uniform random sampling used by Li et al. (2025)).
    - Check that the final two tokens are closing parentheses.
    - Check that the sequence minus the final two tokens would be classified TRUE.
    - If both conditions are met and the sequence has not already been generated, add the sequence to the set of generated sequences.

- Minimizing samples ("almost equal"):
    - Generate sequences uniformly at random (following the same uniform random sampling used by Li et al. (2025)).
    - Check that the difference in number of left and right parentheses is either $2$ or $-2$.
    - For each step in the Dyck path, check that the cumulative number of steps below the $y = 0$ axis is greater than or equal to the cumulative number of steps above the $y = 0$ axis.
    - If both conditions are met and the sequence has not already been generated, add the sequence to the set of generated sequences.

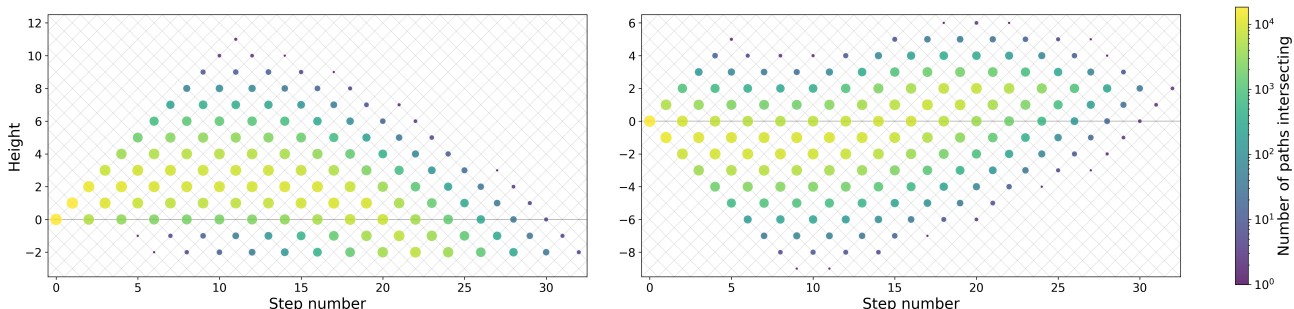

*Figure 19.* Synthetically generated "almost nested" (left) and "almost equal" (right) samples visualized as heatmaps showing the number of Dyck paths crossing each lattice point. "Almost nested" paths climb before returning to near zero (but not zero); "almost equal" paths oscillate around zero height and end near but not at zero. Recall that we identify low OOD accuracy solutions with `Equal-Count` and high OOD accuracy solutions with `Nested`.

Figure 18 gives toy examples of "almost nested" and "almost equal" samples.

The original training distribution from Li et al. (2025) consists of 200k samples: 100k TRUE (balanced and equal count) and 100k FALSE (unbalanced or unequal count), i.e., a 50/50 split, with models trained on 5 epochs of the data. We then use the above criteria to produce the following modified datasets:

- `Almost Nested`: We generate 18.3k "almost nested" samples, then take the original training distribution, (uniformly randomly) remove 36.6k of the original FALSE samples, and then add two copies of each of the 18.3k synthetic "almost nested" samples.

- `Almost Equal`: We generate 19.0k "almost equal" samples, then take the original training distribution, (uniformly randomly) remove 67.1k of the original FALSE samples, and then add four copies of each of the 19.0k synthetic "almost equal" samples.

Figure 19 visualizes the synthetically generated "almost nested" and "almost equal" samples that were used to create `Almost Nested` and `Almost Equal`.

Incidentally, when 100k samples are randomly generated, around 500 of them typically fit the "almost nested" criteria while around 3700 of them fit the "almost equal" criteria, which is evidence that the larger effect size that `Almost Nested` has on the resulting OOD accuracy distribution is related to the relative rarity of "almost nested" samples in the pretraining data.

## C. Implementation Details

### C.1. SGLD Hyperparameters

**Language Modeling.** In the language modeling experiments, we use similar hyperparameters as Wang et al. (2025), with the exception of reducing $\varepsilon$ and increasing the number of draws per chain:

- $n\beta = 30$

- $\gamma = 300$

- $\varepsilon = 3e-4$

- 4 chains

- 300 draws

We reduce $\varepsilon$ and increase the number of draws by a half order of magnitude each. This is a more compute-intensive option, but from a theoretical perspective is unproblematic: we are essentially sampling from the same process with higher time resolution.

**Parentheses Balancing.** We calibrate our SGLD hyperparameters for parentheses balancing experiments to be:

- $n\beta = 100$

- $\gamma = 500$

- $\varepsilon = 3e - 6$

- 4 chains

- 5000 draws

We found that with higher $\varepsilon$ values, it was easier for the sampling process to become unstable, so we opted for a smaller value with many more draws.

### C.2. Scaling Susceptibilities

Susceptibilities estimation, for a given set of model weights and a given set of data samples to measure the susceptibilities of, involves the following computational costs for each SGLD sample:

- A backward pass using the data samples used to compute the gradient for SGLD

- A series of forward passes using each of the data samples being measured

Although backward passes are individually more expensive, if the number of samples being measured in the forward pass is much larger than the number of samples used to compute the SGLD step, then the compute cost is still dominated by the number of data samples being measured, and this is the case in practice. Therefore, we measure the computational cost of our experiments in numbers of forward passes on samples, treating a backward pass as 3 forward passes.

- Training the small language model itself costs around 15m forward passes.

- Computing the susceptibilities for a single model checkpoint's UMAP visualization costs around 40m forward passes (approx. 3x training).

- Computing enough susceptibilities to conduct the induction patterning experiment costs around 7.7b forward passes (approx. 500x training).

Although these experiments are significantly more expensive than a training run for the small language model, this methodology scales favorably with model size. In practice, the LLCs of models have been measured at the billion parameter scale by Urdshals et al. (2025), and susceptibilities have been measured in similar models in preliminary results, so the basic measurements are feasible at scale.

We have reason to believe that the patterning methodology scales favorably as well. As model size increases, optimal-compute training costs increase quadratically, while the cost of computing susceptibilities for a given checkpoint, for a fixed number of weight restrictions, grows linearly. Naively, the number of weight restrictions needed also grows, though in open source models like the Pythia suite, the number of weight restrictions to consider grows much more slowly than quadratically in parameter count. We also find that we can for example exclude many attention heads and still achieve similar results, even in the small language model.

For experiments with significantly higher relative compute costs, such as the induction patterning experiment, we expect more sophisticated approaches to work, such as training an auxiliary model to predict the susceptibilities of a sample.

We also aim to develop the methodology into one that dynamically adjusts the training distribution, rather than computing many susceptibilities up front and applying a fixed distribution shift. This may allow for subtler changes to the distribution, requiring less compute overall. Finally, we may eventually be able to selectively apply patterning to small critical ranges of training rather than to all training steps. Once the weights have been nudged far enough towards a desired structure, the original training distribution may be sufficient for finishing the structure's development. Structural development in language models also appears to occur on a logarithmic time scale, so such nudges may similarly occur on a logarithmic schedule, further reducing compute cost compared to training.

# D. Alignment Applications

Patterning has potential applications to AI alignment. The core insight is that alignment fundamentally concerns the relationship between data, internal structure, and generalization (Lehalleur et al., 2025). Current alignment techniques – RLHF, constitutional AI, safety fine-tuning – operate by shaping the training data distribution, thereby indirectly controlling model behavior. Patterning offers a path toward more direct control: rather than shaping outputs and hoping the right internal structure follows, we target internal structure directly via susceptibilities.

**Simplicity bias and misalignment.** The extension of singular learning theory to reinforcement learning (Elliott et al., 2026) provides a theoretical framework for understanding when misaligned policies may be preferred. The key insight is that a *more optimal policy is not always more optimal from a Bayesian perspective*: the posterior can prefer a simpler policy with higher regret over a complex policy with lower regret.

This may explain goal misgeneralization: the posterior may favor simpler policies capturing spurious correlations ("go to the corner" in an environment where the agent is rewarded for obtaining cheese which is often, but not always, in the corner) over complex policies representing the intended goal ("go to the cheese"). Similar logic may apply to instrumental convergence: acquiring resources or control is a commonly represented pattern across diverse environments, so these behaviors may be simpler to represent than task-specific optimal policies, and hence preferred by the posterior even at higher regret. The same mechanism is relevant to reward hacking: one hypothesis is that reward models learn simple proxies (length, formatting, sycophancy) in part because the region of parameter space implementing "longer responses are better" has lower complexity than the region implementing nuanced quality assessment.

Patterning can potentially address these problems, by raising the complexity of undesirable solutions. By measuring susceptibilities to competing policies, we can identify which training samples differentially affect their complexity, and re-weight accordingly. The parentheses balancing experiments of Section 4 demonstrate this principle in a very simple setting: we identified samples where the susceptibility gap between competing algorithms was large, and used them to steer which solution the posterior favors.

**Constraining undesirable structure.** If instrumental goals like power-seeking correspond to identifiable structures in the model (detectable via susceptibilities to relevant data patterns) then patterning provides a tool for constraining their formation. The key constraint is: *the model should not respond to this pattern*. Formally, this becomes a target $d\mu_{\text{target}}^{\infty}$ in the language of posterior expectation values, and the fundamental equation yields the data intervention that enforces it.

**The specification problem.** These applications presuppose we can specify which structures are desirable. This is nontrivial, but perhaps more tractable than specifying good behavior via training data alone. The language of posterior constraints provides a vocabulary: illustrative examples include "computations should not differ significantly between these distributions" (robustness) and "no component should respond to deception patterns in the data" (honesty). Specification can also be informed empirically by studying misalignment instances through susceptibilities and compiling structural signatures of undesirable generalization into constraints.

**Why mode structure matters.** Susceptibilities measure how a model's internal components respond to patterns in the data distribution (Gordon et al., 2026). These responses develop over training: initially undifferentiated, they become increasingly specialized as the model learns to distinguish different modes (Wang et al., 2025). The key insight is that a single mode manifests across many individual examples in the training data. This means that in general it may be nontrivial to "remove" a mode from the data distribution by naive filtering. Without a grip on mode structure, attempts to eliminate undesirable patterns may fail or have unintended consequences on other modes.

Susceptibility analysis provides this grip. By decomposing the model's responses in terms of modes, we can identify which data perturbations target specific patterns while leaving others unchanged. The fundamental equation $dh_{\text{opt}} = \chi^{\dagger} d\mu_{\text{target}}^{\infty}$ computes the minimum-norm intervention that achieves a desired structural change – potentially allowing for the ability to surgically modify the model's response to some modes without disrupting the rest.

We emphasize that these applications are prospective. Our experiments demonstrate that patterning works, but scaling to frontier models and establishing the full chain from susceptibilities to safety outcomes requires further investigation.

# E. Additional Related Work

**Developmental biology.** The use of *morphogens* to guide organism development provides a useful analogy for patterning. In developmental biology, cell types are characterized by gene expression profiles, and differentiation is modeled as descent through a "developmental landscape" toward distinct cell fates (Waddington, 1957). Mathematically, this is often formalized via bifurcation theory (Rand et al., 2021). A key insight is that small changes in a cell's chemical environment, made at the right time, can have large effects on cell fate. Patterning applies the same principle to neural networks: controlled changes in the data distribution, made at the right point in training, can steer the configuration of the final network. Wang et al. (2025) used susceptibility-based visualizations to study network development as a kind of *embryology*, with the layout of token types on the UMAP playing the role of a body plan.

**Coherent control.** The use of linear response to shape structure is well known in physics, especially in *coherent control* (Brumer and Shapiro, 1989; Shapiro and Brumer, 2012): one first identifies resonant or highly sensitive modes of a molecule or material via low-intensity spectroscopy, then drives those same modes strongly with amplified laser pulses. Other areas of physics revolve around similar "test/push" loops. For instance in condensed matter or femtochemistry, one "tickles" a system with a mild pulse to see which excitations respond, then uses high-intensity pulses in precisely those frequency channels to trigger large-scale changes (e.g. breaking bonds). Patterning follows the same logic: susceptibilities identify which data perturbations the posterior is sensitive to, and we then apply those perturbations to steer internal structure.

# F. Limitations

Our experiments use small models (3M parameters) and simple tasks. Susceptibility estimation is computationally expensive (see Appendix C.2), and scaling to larger models will require more efficient methods. We have focused on one-off interventions made at the start of training, but the full promise of patterning lies in online control (adjusting the data distribution dynamically as training progresses and structure emerges). The theoretical framework assumes infinitesimal perturbations, while practical interventions are necessarily finite; understanding when the linear approximation breaks down remains important. Finally, while we have shown that susceptibility-guided interventions produce predictable changes in observables, the connection between these observables and downstream behavior (e.g. safety properties) requires further study.

