# OpenReview forum: "Patterning: The Dual of Interpretability"
_ICML.cc/2026/Conference — ICML 2026 regular_

### Official Review · Reviewer_JVEG · 2026-02-21

**Soundness:** 2
**Presentation:** 2
**Significance:** 2
**Originality:** 2
**Overall Recommendation:** 3
**Confidence:** 3

**Summary:**

The paper introduces "patterning," a concept framed as the dual problem to mechanistic interpretability. While interpretability seeks to reverse-engineer trained models to understand how they generalize , patterning attempts to determine the specific training data interventions required to produce a desired form of generalization or internal structural configuration.

The methodology is grounded in Structural Bayesianism and Singular Learning Theory. The authors utilize susceptibilities, which quantify how the posterior expectation values of specific structural observables respond to infinitesimal shifts in the training data distribution. By modeling this relationship as a linear response system, the authors derive the fundamental equation of patterning. This equation leverages the Moore-Penrose pseudoinverse of the susceptibility matrix to map a targeted change in a model's internal structure to the optimal, minimum-norm intervention required in the training data.

The authors empirically validate this framework through two primary experimental settings. First, they target the induction circuit in a small language model. By re-weighting the training data along the second principal component of the susceptibility matrix, which couples induction patterns in the data to the induction circuit in the weights, they demonstrate the ability to predictably accelerate, delay, or entirely prevent the formation of induction heads. Second, the authors apply patterning to a synthetic parenthesis balancing task where two distinct algorithms, Nested and Equal-Count, both achieve perfect training accuracy. By targeting the local learning coefficient, which measures the complexity of a solution basin, the authors use susceptibility gaps to identify highly discriminating training samples. Up-weighting these specific samples successfully raises the complexity of the undesired solution, reliably steering the posterior to favor the targeted algorithm.

**Compliance With Llm Reviewing Policy:**

Affirmed.

**Final Justification:**

The authors’ rebuttal addressed the technical questions well. However, my concerns regarding the paper’s overall originality and significance remain. The novelty of the proposed method is still incremental, and its impact on the field seems limited. While the technical clarifications are appreciated, they do not change my fundamental assessment of the work's contribution. I am maintaining my score.

**Key Questions For Authors:**

1. In Section 4.4, the application of the "Almost Equal" synthetic dataset unexpectedly decreased the local learning coefficient (LLC) across all solutions, rather than selectively raising it for the Equal-Count algorithm as intended. The submission attributes this discrepancy to the breakdown of the linear response approximation under finite perturbations. How can practitioners reliably predict, bound, or correct for these non-linear effects when designing real-world data interventions? Providing a theoretical bound or a practical heuristic to test for finite perturbation breakdown before committing to a full retraining run would helpful.

2. There appears to be a gap between the dynamic nature of the theoretical framework and the static nature of the experimental application. The theory models infinitesimal perturbations based on the posterior at a specific state. However, in the induction circuit experiments, susceptibilities are measured around step 4,000, yet the resulting fixed token mask is applied statically across multiple epochs up to step 20,000. How sensitive is the patterning intervention to the specific timing of the susceptibility measurement, and does the optimal data re-weighting drift as the model's internal structures evolve?

3. Appendix A.4 reveals that re-weighting based on induction patterns causes unintended "collateral damage" to other internal structures, visibly altering the susceptibilities for Word Part, Word End, and Numeric patterns. Does the fundamental equation of patterning ($dh_{opt} = \chi^\dagger d\mu_{target}^\infty$) mathematically accommodate explicit constraints to hold other specific structural coordinates constant (e.g., explicitly setting $d\mu_{other} = 0$ in the target vector)?

4. The computational cost of estimating susceptibilities via Stochastic Gradient Langevin Dynamics (SGLD) is a critical limitation, requiring approximately 7.7 billion forward passes just to conduct the induction patterning experiment on a 3M parameter model. While Appendix C.2 suggests future optimizations such as training auxiliary predictive models or applying patterning only during critical training windows, is there any preliminary empirical evidence or theoretical justification indicating that the required number of SGLD samples grows sub-linearly with model size?

**Limitations:**

Yes

**Strengths And Weaknesses:**

The paper presents a highly original and conceptually elegant framework by introducing "patterning" as the dual to mechanistic interpretability. While much of the field focuses on reverse-engineering existing structures, this work takes a significant step forward by using Singular Learning Theory (SLT) and susceptibilities to proactively "write" internal structures through principled data curation. This shift from passive observation to active structural control has profound implications for machine learning, particularly for AI alignment and ensuring safe out-of-distribution generalization, making the work highly significant.

The submission is technically sound, and its core claims are well-supported by carefully designed empirical experiments. The mathematical formulation, specifically using the Moore-Penrose pseudoinverse to invert the linear response relationship, is logically sound and well-articulated. Furthermore, the experimental settings—modulating the induction circuit in a small language model and selecting between the Nested and Equal-Count algorithms in a parenthesis balancing task—clearly validate the theoretical predictions. The authors successfully demonstrate that targeting specific susceptibilities and local learning coefficients can predictably steer model development and algorithm selection.

However, the paper's soundness is somewhat hindered by the limitations of the linear response approximation when applied to finite data perturbations. The authors transparently acknowledge this limitation when discussing the "Almost Equal" dataset, which unexpectedly lowered the local learning coefficient for all solutions rather than selectively raising it for the Equal-Count algorithm as intended. Additionally, the computational cost of estimating susceptibilities via Stochastic Gradient Langevin Dynamics (SGLD) is extremely high, requiring approximately 7.7 billion forward passes just to conduct the induction patterning experiment on a 3M parameter model. While the theoretical foundations are strong, this computational burden raises valid concerns regarding the immediate scalability of the method to frontier models.

Regarding presentation, the manuscript is exceptionally clear and well-structured. The overarching narrative is easy to follow, successfully bridging abstract theoretical concepts from SLT with concrete, observable phenomena in neural networks. The visual aids, particularly the UMAP embeddings showing the formation and stunting of organizational "fins" and the Dyck path representations for the parenthesis task, significantly enhance the reader's understanding of the mechanisms at play. Finally, the authors effectively position their work within the broader context of existing literature, clearly distinguishing patterning from traditional influence functions and standard data curation techniques.

---

> ### Author Rebuttal · Authors · 2026-03-30
>
> Thanks for your review, for your questions, and for your very positive comments on the originality and significance of the work and its presentation. We appreciate the chance to clarify our results.
>
> Regarding the failure of linear approximation discussed in section 4.4: this discussion is written conflating the idea that if the susceptibility gap is large, it must be the case that the solution being selected against has its LLC increased (“We designed these samples to raise the LLC at Equal-Count solutions”). However, the susceptibility gap can in principle be maximized by samples where the solution being selected against has a neutral response to the samples, while an alternate solution has its LLC significantly reduced. In this case, the samples would have near-zero susceptibilities on the Equal-Count solution models, while having negative susceptibilities for the Almost-Equal solutions. This would then have the exact effect seen in the paper, where the LLCs are *differentially* affected, but not necessarily by *increasing* one of the LLCs, and which would still have the intended steering effect.
>
> We re-examined our susceptibility data after the paper submission and noticed this was in fact the case: the samples we used to shift the data distribution to favor Nested solutions did have negative susceptibilities for Almost-Equal solutions and closer to neutral susceptibilities for Equal-Count solutions. Consequently, there is no breakdown of the linear approximation after all. We will correct this detail in the camera-ready version if accepted.
>
> Regarding the concern about scalability: yes, there is ample preliminary evidence that the cost of compute does scale significantly sub-linearly with model scale. We already find this methodology computationally tractable in larger models up to the billion parameter scale in preliminary work. Additionally, we find that with a modest amount of training data on estimated susceptibilities, we can indeed train an auxiliary model to predict susceptibility values with enough accuracy that we can run experiments on them as a drop-in replacement. This reduces the computational cost by orders of magnitude, and additional progress can be made in this direction. Additionally, the type of intervention we consider here is one which is amenable to deriving for a smaller model and transferring to a larger one, since the training data reweighting itself may be directly re-applied to another model. We therefore do not expect that compute limitations will be an ultimate blocker for applications to larger models.
>
> Regarding the sensitivity of the intervention to the timing of measuring susceptibilities: we do not measure the susceptibilities at training step 4,000, rather we measure the susceptibilities at the end of training for enough data samples to *cover* 4,000 training steps. We then train on multiple epochs up to step 20,000. We do not believe this choice should have a significant impact, but we compare with the baseline case of no reweighting to ensure that there is a fair apples to apples comparison. Ultimately we advocate for the development of this technique into something that is done continuously throughout training, which would address this concern. However, we argue this is still justified when only using end-of-training susceptibilities. Even though the internal representations (i.e. activations, embeddings) may shift and rotate throughout training, the underlying pattern in the data itself remains static. The susceptibilities translate from the model internals to the more static pattern in the data, therefore it should be possible in principle to extract a similar data reweighting out of any model checkpoint that has already developed the induction circuit. In this sense, the methodology may be more robust to the timing of measurements than more established techniques such as activation steering.
>
> Regarding the unintended “collateral damage”: yes, the fundamental equation of patterning does accommodate constraints that would hold other structural coordinates constant. Part of the motivation for using the Moore-Penrose pseudo-inverse is that this is the minimum norm intervention that aims at the structural target, so mathematically this is the smallest and most surgical intervention possible. In practice, the noted “collateral damage” must be relatively limited in impact: we compare the overall loss curves of the various interventions and we find that the various interventions remain relatively well-coupled in their overall training loss trajectory. In particular, patterns like Word Part or Word End must be fairly large patterns in the data, and so significant damage to the model’s ability to handle these patterns would be very clearly visible in the loss curves.
>
> We hope that these address your concerns about the paper, and that given your otherwise positive impressions of the originality, significance, and presentation, that you will consider raising your score.

---

> > ### Author Rebuttal · Reviewer_JVEG · 2026-04-01
> >
> > Thank you sincerely for taking the time to prepare such a thoughtful rebuttal. I appreciate the effort the authors have put into addressing the concerns raised. I will be maintaining my score.

---

### Official Review · Reviewer_8L7z · 2026-03-09

**Soundness:** 2
**Presentation:** 3
**Significance:** 2
**Originality:** 3
**Overall Recommendation:** 4
**Confidence:** 4

**Summary:**

This work studies the patterning problem: how to induce a target generalization behavior by intervening on training data. The authors proposed a susceptibility-guided method grounded in SLT and prior work on susceptibilities. The proposed method is empirically verified on two toy tasks: Induction circuit and parenthesis balancing task. On the two toy tasks, the proposed method can effectively induce different generalization behaviors by reweighting the training tokens.

**Compliance With Llm Reviewing Policy:**

Affirmed.

**Final Justification:**

I appreciate the authors for their response. I agree with other reviewers that this work is original and exciting, especially from a theoretical point of view. However, much of the excitement has not been empirically verified on realistic model/task settings -- especially given there are many mathematically appealing framework in the neighboring predictive data attribution area, yet very few of them have proven to be useful. Overall, I will increase my rating to 4.

**Key Questions For Authors:**

Please address the weaknesses above. In particular, I am curious about what are the advantages of the proposed direction, i.e., model weights => training data =>  target generalization behavior vs. the current workflow widely used in steering/editing, i.e., model weights/representations => target generalization behavior. What benefits do we get from intervening on the training data?

**Limitations:**

Yes.

**Strengths And Weaknesses:**

**Strengths**
- This work proposes a new method to control data distribution to induce a specific generalization behavior that is theoretically grounded in the SLT and susceptibilities. While this is not a new direction (see weaknesses), this direction is relatively understudied, and it is worth to explore more methods in this space.
- The proposed method is empirically validated on two toy tasks with well-studied internal structures, where the results show intended effects on generalization.
- The authors did a good job on providing the necessary background knowledge on SLT and susceptibilities, making this work more accessible to audience who are not familiar with this line of work.

**Weaknesses**:
- From a evaluation point of view, no baseline has been compared in this work. If the ultimate goal is to control or induce certain generalization behaviors, then we should compare with methods that can achieve the same goal. This could include (but not limited to) predictive data attribution methods that use gradient/metagraident, where the attribution scores can be used for data selection. More broadly speaking, this could also include methods that induce generalization behaviors with interventions on model representations bypassing the training data. What is the advantage of modulating through data vs. modulating through internal representation (e.g., steering, model editing)? It would be great if the authors could provide some empirical comparisons.
- Unclear how the method perform on more complicated tasks that involve model components beyond attention heads
  - The two toy tasks are clean and simple compared with downstream tasks typically used for evaluating a pre-trained networks. In both case, the key model components that mediate the desired generalization behavior are a small set of attention heads. For more complicated tasks, model components might interact with each other, e.g., earlier layer MLP have indirect effects on later layer MLP. Would the proposed method still be effective in such cases? How do know if there is any unintended effect of the data interventions?
  - If I understand correctly, the bucket balancing case assumes we already have access to the weights of the two solutions. How would the proposed method work when we only have a target generalization behavior (defined by an OOD test set)?
- Applicability to modern LLMs: The theoretical results from SLT and susceptibilities assume convergence, i.e., local minima, yet for modern LLMs, training typically stops before convergence. To what extent does the proposed intervention remain effective on models that are not trained to convergence?
- The idea of using data distribution to modulate generalization behaviors has been explored in the literature, e.g., in the context of ICL (Chan et al. 2022, Singh et al. 2024), induction heads (Singh et al. 2024) linear representations (Merullo et al. 2024), and more broadly in the data attribution/selection literature (Ilyas et al. 2025, Nadkarni et al. 2025). Despite this work coining a new term "patterning", the underlying problem has been studied in the literature.
- Calling patterning the "Dual of Interpretability" is an interesting choice, which inevitably draws the boundary of what counts as "interpretability" work that the community may or may not agree. Maybe it's better to directly define what patterning is in the title, as opposed to define it vaguely as the opposite of another ambiguous term "Interpretability".

---

> ### Author Rebuttal · Authors · 2026-03-30
>
> Thanks for your review, and for your questions. We appreciate the chance to clarify our results.
>
> Regarding moving beyond the use of attention heads: in principle this methodology can use MLPs as model components, and in fact permits arbitrary weight subspaces, including non-parameter-aligned ones, though that might be unnatural for other reasons.
>
> Regarding unintended effects of the data interventions: it is certainly possible, however the pseudo-inverse in the fundamental equation of patterning is the minimum-norm intervention by definition, which should be as surgical as possible.
>
> Regarding the reliance on access to two solutions: in the course of typical model training, you may think of two different checkpoints, A and B, as being your two models to compare. If checkpoint B then moves in some direction you either prefer or disprefer compared to A, you can then similarly find samples in the training data that steer further training towards or away from B relative to A using this methodology.
>
> Regarding effectiveness on models not trained to convergence: it is true that the theoretical hypotheses for applicability of our methods (including LLC estimation and susceptibility estimation) require sampling to be near a local minima of the population loss, and this may not be true in practice. One response (in “Modes of sequence models and learning coefficients”) posits that at a given resolution (set by the stepsize and inverse temperature used for SGLD sampling) the SGLD process returns information about an “effective potential” for which the model is at a local minima. However, this hypothesis has not been studied extensively.
>
> Regarding prior work using the data distribution to modulate generalization: we appreciate the comparison and agree that the relationship between data distribution and learned structure is studied in prior work. However, we believe there is a meaningful distinction between (i) observational studies that show distributional properties correlate with emergent behaviors, (ii) data attribution methods that optimize scalar performance metrics, and (iii) patterning, which provides an invertible quantitative framework for computing data interventions that target internal structural coordinates.
>
> Chan et al. (2022) and Singh et al. (2024) demonstrate that coarse-grained distributional properties affect whether ICL or specific circuits emerge. Singh et al. introduce an "artificial optogenetics" framework that performs causal interventions on activations during training (e.g., clamping a layer's attention pattern) to identify which subcircuits bottleneck induction head formation. This is a form of causal analysis of circuit development, but the interventions are on model internals, not on data: it answers "which subcircuit is the bottleneck?" rather than "which training samples should be reweighted to accelerate or delay this circuit?" Patterning addresses the latter question. Merullo et al. (2025) establish a correlation between pretraining frequency and linear representation quality, but perform no data interventions at all as far as we understand the work.
>
> Ilyas et al. (2025) and Nadkarni et al. (2025) do provide principled frameworks for data selection/attribution, but their targets are scalar performance metrics (test loss, factual recall). Our Section 4 demonstrates steering between two algorithms that achieve identical training and in-distribution test loss, by targeting structural coordinates (local learning coefficients) rather than behavioral outcomes.
>
> The term "patterning" is introduced not to rebrand data selection, but to name a specific problem (targeting internal structure rather than behavior) solved by a specific method (inverting the susceptibility matrix). We appreciate the question and references and will add them along with a version of the above to the related work section.
>
> Regarding the advantages of our proposed direction vs. the more widely used steering workflow, and the benefits of intervening on the training data: first, these two approaches may be highly complementary and composable rather than being in competition. Data-level interventions may also be more robust than activation level ones. If a capability exists somewhere in the model weights, it is always possible in principle to bypass activation steering and elicit them, whereas data-level interventions may entirely erase those capabilities; they are “deeper” in this sense. Finally, data interventions are more transferrable: activation space may change dramatically between different models, while data reweightings are more agnostic, as they target static patterns in the data itself. There are more advantages we could highlight, but in short, we think these are quite distinct as approaches, and neither comes close to pareto dominating the other.

---

> > ### Author Rebuttal · Reviewer_8L7z · 2026-04-03
> >
> > I appreciate the authors for their response. I agree with other reviewers that this work is original and exciting, especially from a theoretical point of view. However, much of the excitement has not been empirically verified on realistic model/task settings -- especially given there are many mathematically appealing framework in the neighboring predictive data attribution area, yet very few of them have proven to be useful. Overall, I will increase my rating to 4.

---

### Official Review · Reviewer_NMHt · 2026-03-10

**Soundness:** 4
**Presentation:** 3
**Significance:** 4
**Originality:** 4
**Overall Recommendation:** 5
**Confidence:** 4

**Summary:**

The paper introduces the concept of patterning to answer the question of what is the effect of infinitesimal changes in the data on the specific observables. To do so, they resort to susceptibilities, a tool grounded in singular learning theory which gives a linear relationship between data and observables. By inverting this relationship, they can shift the data distribution for the parameter posterior to concentrate around areas which give rise to a desired expected observable. They apply this methodology to two scenarios. In the first application they successfully reinforce the formation of induction circuits in small language models. In the second application, they consider a model which classifies strings of parentheses as correct or incorrect. There are at least two distinct generalising solutions and they manage to use susceptibilities to steer the learning algorithm into a specific generalising solution.

**Compliance With Llm Reviewing Policy:**

Affirmed.

**Final Justification:**

The rebuttal addressed some of the minor concerns I had about the paper, reinforcing my positive score. Again, it is a technically sound paper, which uses a very original SLT-approach to explore the problem of understanding the effect of shifts in data distributions to the internal configuration of a model. It is exciting to see what research stems from this paper!

**Key Questions For Authors:**

Most have been already discussed above. An extra question which is pertinent with the current research landscape is the scalability question. Deriving re-weighting of training samples is a great method, but how feasible is this for larger models. Computing LLCs is expensive and minimising solutions may not be known a priori. How can these methods be extended for training new models. If scalability is a problem, how can LLM researchers see value in these methods?

**Limitations:**

Great summary of potential limitations discussed in appendix F. It is briefly mentioned in a footnote but, as with other aspects of SLT, it is still unclear how the Bayesian learning framework can be applied to SGD-based methods. A more detailed discussion in an appendix or in future work would be welcomed.

**Strengths And Weaknesses:**

General impression:
Overall, this is an incredibly interesting, exciting and potentially useful research paper which opens up further avenues of research for interpretability and alignment. It is exciting to see new theory coming from SLT being applied to successfully. There are concerns about scalability and applicability, but this should only motivate more researchers to engage with this area of research.

Strengths and weaknesses:
1. The paper is very sound, building upon previous work on susceptibilities and SLT more generally. The theoretical foundations are strong and complete, with details relegated to the appendices when appropriate (e.g. appendix B.1). The paper also conducts two diverse experiments which tackle two different aspects. The induction circuit experiment can be framed within interpretability research, whereas the parenthesis balancing task seems to be more of an alignment experiment where a different generalising solution can be favoured by rebalancing the data. Some concerns:
   1.a. I would be interested in analysing the sensitivity of the SGLD estimates with respect to the hyperparameters. It seems like other similar SGLD estimators like the LLC suffer from high hyperparameter sensibility. It is only briefly discussed how the hyperparameters were chosen, but it would be interesting to know more. Are there serious changes in the estimates? Is this problematic, or are they changes in the scale of the estimates but the trends observed as still respected?
2. The paper is well-structured and complete. It provides a new method to understand the relationship between a model's internal structure and its training data, and provides two non-trivial experiments to corroborate the theory developed. In the second experiment, the results don't seem to align completely with the expectations, but the authors are honest and provide plausible explanations for this divergence with theoretical predictions. This allows the reader to better comprehend the limitations and potential directions for further research. Some concerns:
   2.a. There seems to be three notions of susceptibility discussed: (i) equation 2, i.e. $d\mu^\infty =\chi dh$; (ii)  the definition of susceptibility as a derivative as presented in equation 14; and (iii) definition 2.1 The three notions are partially reconciled after definition 2.1, but it still feels unclear. Indeed, the mixture of densities example in lines 188-192 is a good idea but ends up being confusing due to notational abuse of $\chi$.
3. Development of the novel theoretical framework of patterning, based on the recent concept of susceptibilities, continuing to build up the SLT machinery and furthering the understanding of susceptibilities. It opens the doors to more theoretically grounded work on alignment and interpretability, which is extremely significant in current ML research.
4. The paper is extremely original. It builds up theoretical and practical tools for interpretability and alignment. It is not an incremental change, it a novel framework within SLT which hopefully inspires more people to pursue further research in this area.
5. Will the code be made public?
Small typos:
1. Incorrect font for Equal-Count in line 321, column 2.

---

> ### Author Rebuttal · Authors · 2026-03-30
>
> Thanks for your review, for your questions, and for your very positive comments about your general impressions.
>
> Regarding the sensitivity of estimates with respect to hyperparameters: SGLD hyperparameter selection remains a fairly tricky part of the methodology. In the case of the LLC, the process for selecting hyperparameters takes into account this sensitivity and selects regions of hyperparameter space where estimates are relatively more stable (for example in the observed trends), and where other assumptions about what a “healthy” sampling process looks like continue to hold. Some examples might include the specific expected behavior as step size is varied, or whether a region of hyperparameter space becomes insensitive to changes in certain hyperparameters, sometimes in specific ways (suggesting that the effect of other hyperparameters may be “dominating” in that region, in a theoretically undesirable way). The LLC estimator is relatively simple in that it is, up to scaling and translation, just the expectation of the loss over the posterior, so it is arguably the simplest single proxy for how well the posterior is being sampled. In practice, our methodology then relies on using the same process for hyperparameter selection, focusing on LLCs rather than susceptibilities in the calibration process. We don’t expect this to be problematic, but we agree that it would be valuable to eventually integrate our understanding of susceptibilities into this calibration process as well.
>
> Regarding the three notions of susceptibility: we agree this is confusing as written. The intended presentation is to start with (1) the susceptibility as a rate of change of an expectation value and then (2) present the general susceptibility as an integral of a susceptibility density (the per-token susceptibility). But this is muddled by the partial involvement of the component C in the per-token susceptibility but not the general definition, and the terse treatment of 188-192 as you note. If accepted we’ll use the extra page in part to decompress this and give it the space it deserves to communicate clearly. Thanks for the feedback\!
>
> Regarding the code being made public: we plan to open source a demo notebook that replicates the parentheses balancing experiments. The induction circuit patterning experiments rely on experimental infrastructure that would be difficult to open source.
>
> Regarding the concern about scalability: we already find this computationally tractable in larger models up to the billion parameter scale in preliminary work. Additionally, we find that with a reasonable but still relatively modest amount of training data on estimated susceptibilities, we can train an auxiliary model to predict susceptibility values with enough accuracy that we can run experiments on them as a drop-in replacement. These predicted values are not perfect, but we think additional progress can be made in this direction, and this approach reduces the computational cost by orders of magnitude. Additionally, the type of intervention we consider here is one which is amenable to deriving for a smaller model and transferring to a larger one, since the training data reweighting itself may be directly re-applied to another model.
>
> We are therefore cautiously optimistic that compute limitations will not be an ultimate blocker for applications to LLMs. One way this could be false is if larger models require orders of magnitude more SGLD steps than small models in order to get sufficiently accurate susceptibilities to pattern. However we note that in our current methodology the per-component posteriors are over parameter spaces that scale with the size of the component (not the whole model) and sampling for different components can be done in parallel, so this does not seem likely to be a fundamental blocker.

---

> > ### Author Rebuttal · Reviewer_NMHt · 2026-04-01
> >
> > Thank you for taking the time to write such a detailed response. I am specially happy about:
> >
> > (i) You intend to use the extra space to clarify the different notions of susceptibilities which were introduced in the paper.
> >
> > (ii) You intend to release _some_ of the code for other researchers to explore this.
> >
> > (iii) The response to scalability is specially interesting. I would be very intrigued in seeing how auxiliary models can be used to predict susceptibilities, and whether this can be transferred to larger models.
> >
> > Well done for this paper! It's truly original and exciting.

---

### Official Review · Reviewer_ifdV · 2026-03-12

**Soundness:** 3
**Presentation:** 3
**Significance:** 2
**Originality:** 4
**Overall Recommendation:** 4
**Confidence:** 3

**Summary:**

This paper proposes "patterning" as the reverse of mechanistic interpretability. Instead of looking inside a trained network to understand what structures it learned, patterning asks: if I want a specific internal structure, what training data should I use? The method relies on susceptibilities, which measure how sensitive the model's internal organization is to changes in the training data. These form a linear relationship between data changes and structural changes, and you can invert that relationship (via a pseudoinverse) to get a recipe for how to adjust the data. They test this in two experiments. First, they speed up or prevent the formation of the induction circuit in a small language model by re-weighting training tokens. Second, in a parenthesis matching task where two algorithms both get perfect training accuracy, they steer which algorithm the model learns by adding carefully chosen synthetic samples. The first experiment works cleanly. The second works directionally but one of the two interventions operates through a completely different mechanism than the theory says it should.

**Compliance With Llm Reviewing Policy:**

Affirmed.

**Key Questions For Authors:**

For the Almost Equal intervention, did you consider sweeping over perturbation sizes (e.g. gradually increasing the number of synthetic samples added)? If the linear approximation is the issue, small perturbations should produce LLC changes in the predicted direction, with deviations appearing at some crossover point. This would let you map where the linear regime actually ends. Is there a practical reason this was not done?

**Limitations:**

Yes, they are upfront about the scaling limitations and about the result that contradicts their theory.

**Strengths And Weaknesses:**

Strengths:
The framing is very original. Flipping the susceptibility framework from reading internal structure to writing it is, as far as I know, genuinely new. The combination of singular learning theory, Bayesian inference, and data curation into a single framework feels like more than the sum of its parts.

The theory is strong. The derivation from susceptibilities through SVD to the patterning equation is very well done

The experiments are well designed. The induction circuit experiment has a natural dose-response structure with four conditions, and Figure 2 shows exactly the monotonic effect you would expect. Extensive appendix analyses (UMAPs, per-pattern susceptibilities, explained variance, loss curves) show the intervention is fairly surgical. The parenthesis experiment is well designed.

Weaknesses:
The linear approximation fails in one of the paper's own experiments and the failure is not adequately explained. The authors attribute this to finite perturbations exceeding the linear regime, but provide no quantitative analysis of when or why the approximation breaks down, no higher-order corrections, and no way to predict this ahead of time.

This casts doubt on scalability. If the framework gives a qualitatively wrong mechanistic prediction for a 2-by-m matrix on a toy task, it is hard to be confident about larger models with thousands of interacting structures. The scaling discussion in Appendix C.2 focuses on computational cost but does not address whether the mathematical assumptions hold at scale.

---

> ### Author Rebuttal · Authors · 2026-03-30
>
> Thanks for your review, and for your questions. We also appreciate your highlighting the originality of the work.
>
> One major point raised is in one of the two interventions applied to the parentheses balancing setting. In particular, the concern as we understand it is that the linear approximation fails, in the sense that the LLC is lowered rather than raised. We discuss this briefly in section 4.4.
>
> This discussion is written conflating the idea that if the susceptibility gap is large, it must be the case that the solution being selected against has its LLC increased (“We designed these samples to raise the LLC at Equal-Count solutions”). However, it is possible in principle that the susceptibility gap is maximized by samples where the solution being selected against has a neutral response to the samples, while an alternate solution has its LLC significantly reduced. In this case, that would mean that the samples have near-zero susceptibilities on the Equal-Count solution models, while having negative susceptibilities for the Almost-Equal solutions. This would then have the exact effect seen in the paper, where the LLCs are *differentially* affected, but not necessarily by *increasing* one of the LLCs, and which would still have the intended effect (as observed in the paper) of shifting the distribution of solutions.
>
> We re-examined our susceptibility data after the paper submission and noticed this was in fact the case: the samples we used to shift the data distribution to favor Nested solutions did have negative susceptibilities for Almost-Equal solutions and closer to neutral susceptibilities for Equal-Count solutions. Consequently, there is no breakdown of the linear approximation after all. We apologize for the confusion and for missing this detail when writing the original submission, and we will correct this in the camera-ready version if accepted.
>
> Regarding the related concern about scalability: we hope that the correction on the linear approximation point addresses most of the worry here. Additionally, we see in recent experiments that we can in principle run susceptibilities at the billion parameter model scale, though it still empirically remains to be seen whether the mathematical assumptions continue to hold.
>
> Regarding sweeping over perturbation sizes: given the correction above to our analysis on the linear approximation regime, we hope the main concern about a lack of a sweep over perturbation size here is addressed. Still, we believe that a sweep over perturbation sizes would be interesting in any case to characterize the linear approximation regime, and we can add such experiments to the paper.

---

> > ### Author Rebuttal · Reviewer_ifdV · 2026-04-01
> >
> > I thank the authors for their rebuttal. I maintain my score for my now as I believe my concerns have not been addressed concretely with new experiments/results. If the authors can provide some of the preliminary results/extra experiments they mention, I will consider raising my score.

---

> > > ### Author Response · Authors · 2026-04-01
> > >
> > > Thank you for your response. We will try to elaborate on our comments on scalability in two parts. With regards to the practical feasibility of computing susceptibilities at scale, we can point to the recent work in "Compressibility Measures Complexity: Minimum Description Length Meets Singular Learning Theory" by Urdshals et al. That paper studies the LLC, however the difficult part of estimating both LLCs and susceptibilities at scale lies in the underling SGLD sampling process, and so feasibility there translates to feasibility for susceptibilities estimation. With regards to whether the patterning methodology continues to hold at scale, the specific preliminary results that we refer to in our rebuttal above involve a 2B parameter reward model. In that case, we apply a similar methodology as in the parentheses balancing case and observe that the linear response approximation holds for relatively large distribution shifts in that setting. We achieve something like a 30% reduction in the strength of formatting bias without apparent damage to other performance based on initial evaluations.

---

### Decision · Program_Chairs · 2026-04-30

**Decision:**

Accept (regular)

**Comment:**

This paper proposes a reformulation of internal mechanisms as being produced by particular inputs during training using a susceptibility based method.
The reviewers agree that it is a very interesting approach and both the theory and empirical findings are promising.
A large concern is whether these methods can scale to larger models and more realistic settings. I think that the paper is interesting despite this limitation, as scaling can often be performed in followup work. Another concern is that some of the experiments directly contradict the theory, although the authors provide plausible hypotheses as to why they might form an exception to the rule. The issue with this experiment is likely caused by limitations in their approximation, which could be a concern in some realistic settings and is probably the largest issue in the paper.